# Matrix Completion has No Spurious Local Minimum

**Rong Ge**
Duke University
308 Research Drive, NC 27708
rongge@cs.duke.edu.

**Jason D. Lee**
University of Southern California
3670 Trousdale Pkwy, CA 90089
jasonlee@marshall.usc.edu.

**Tengyu Ma**
Princeton University
35 Olden Street, NJ 08540
tengyu@cs.princeton.edu.

## Abstract

Matrix completion is a basic machine learning problem that has wide applications, especially in collaborative filtering and recommender systems. Simple non-convex optimization algorithms are popular and effective in practice. Despite recent progress in proving various non-convex algorithms converge from a good initial point, it remains unclear why random or arbitrary initialization suffices in practice. We prove that the commonly used non-convex objective function for *positive semidefinite* matrix completion has no spurious local minima – all local minima must also be global. Therefore, many popular optimization algorithms such as (stochastic) gradient descent can provably solve positive semidefinite matrix completion with *arbitrary* initialization in polynomial time. The result can be generalized to the setting when the observed entries contain noise. We believe that our main proof strategy can be useful for understanding geometric properties of other statistical problems involving partial or noisy observations.

## 1 Introduction

Matrix completion is the problem of recovering a low rank matrix from partially observed entries. It has been widely used in collaborative filtering and recommender systems [Kor09, RS05], dimension reduction [CLMW11] and multi-class learning [AFSU07]. There has been extensive work on designing efficient algorithms for matrix completion with guarantees. One earlier line of results (see [Rec11, CT10, CR09] and the references therein) rely on convex relaxations. These algorithms achieve strong statistical guarantees, but are quite computationally expensive in practice.

More recently, there has been growing interest in analyzing non-convex algorithms for matrix completion [KMO10, JNS13, Har14, HW14, SL15, ZWL15, CW15]. Let $M \in \mathbb{R}^{d \times d}$ be the target matrix with rank $r \ll d$ that we aim to recover, and let $\Omega = \{(i,j) : M_{i,j} \text{ is observed}\}$ be the set of observed entries. These methods are instantiations of optimization algorithms applied to the objective[1],

$$f(X) = \frac{1}{2} \sum_{(i,j) \in \Omega} \left[ M_{i,j} - (XX^\top)_{i,j} \right]^2, \tag{1.1}$$

These algorithms are much faster than the convex relaxation algorithms, which is crucial for their empirical success in large-scale collaborative filtering applications [Kor09].

All of the theoretical analysis for the nonconvex procedures require careful initialization schemes: the initial point should already be close to optimum. In fact, Sun and Luo [SL15] showed that after this initialization the problem is effectively strongly-convex, hence many different optimization procedures can be analyzed by standard techniques from convex optimization.

However, in practice people typically use a random initialization, which still leads to robust and fast convergence. Why can these practical algorithms find the optimal solution in spite of the non-convexity? In this work we investigate this question and show that the matrix completion objective has *no spurious* local minima. More precisely, we show that any local minimum $X$ of objective function $f(\cdot)$ is also a global minimum with $f(X) = 0$, and recovers the correct low rank matrix $M$.

Our characterization of the structure in the objective function implies that (stochastic) gradient descent from arbitrary starting point converge to a global minimum. This is because gradient descent converges to a local minimum [GHJY15, LSJR16], and every local minimum is also a global minimum.

## 1.1  Main results

Assume the target matrix $M$ is symmetric and each entry of $M$ is observed with probability $p$ independently [2]. We assume $M = ZZ^\top$ for some matrix $Z \in \mathbb{R}^{d \times r}$.

There are two known issues with matrix completion. First, the choice of $Z$ is not unique since $M = (ZR)(ZR)^\top$ for any orthonormal matrix $Z$. Our goal is to find one of these equivalent solutions.

Another issue is that matrix completion is impossible when $M$ is "aligned" with standard basis. For example, when $M$ is the identity matrix in its first $r \times r$ block, we will very likely be observing only 0 entries. To address this issue, we make the following standard assumption:

**Assumption 1.** *For any row $Z_i$ of $Z$, we have $\|Z_i\| \leqslant \mu/\sqrt{d} \cdot \|Z\|_F$. Moreover, $Z$ has a bounded condition number $\sigma_{\max}(Z)/\sigma_{\min}(Z) = \kappa$.*

Throughout this paper we think of $\mu$ and $\kappa$ as small constants, and the sample complexity depends polynomially on these two parameters. Also note that this assumption is independent of the choice of $Z$: all $Z$ such that $ZZ^T = M$ have the same row norms and Frobenius norm.

This assumption is similar to the "incoherence" assumption [CR09]. Our assumption is the same as the one used in analyzing non-convex algorithms [KMO10, SL15].

We enforce $X$ to also satisfy this assumption by a regularizer

$$f(X) = \frac{1}{2} \sum_{(i,j)\in\Omega} \left[ M_{i,j} - (XX^\top)_{i,j} \right]^2 + R(X), \qquad (1.2)$$

where $R(X)$ is a function that penalizes $X$ when one of its rows is too large. See Section 4 and Section A for the precise definition. Our main result shows that in this setting, the regularized objective function has no spurious local minimum:

**Theorem 1.1.** *[Informal] All local minimum of the regularized objective (1.1) satisfy $XX^T = ZZ^T = M$ when $p \geqslant poly(\kappa, r, \mu, \log d)/d$.*

Combined with the results in [GHJY15, LSJR16] (see more discussions in Section 1.2), we have,

**Theorem 1.2** (Informal). *With high probability, stochastic gradient descent on the regularized objective (1.1) will converge to a solution $X$ such that $XX^T = ZZ^T = M$ in polynomial time from any starting point. Gradient descent will converge to such a point with probability 1 from a random starting point.*

Our results are also robust to noise. Even if each entry is corrupted with Gaussian noise of standard deviation $\mu^2\|Z\|_F^2/d$ (comparable to the magnitude of the entry itself!), we can still guarantee that all the local minima satisfy $\|XX^T - ZZ^T\|_F \leqslant \varepsilon$ when $p$ is large enough. See the discussion in Appendix B for results on noisy matrix completion.

Our main technique is to show that every point that satisfies the first and second order necessary conditions for optimality must be a desired solution. To achieve this we use new ideas to analyze the effect of the regularizer and show how it is useful in modifying the first and second order conditions to exclude any spurious local minimum.

## 1.2 Related Work

**Matrix Completion.** The earlier theoretical works on matrix completion analyzed the nuclear norm heuristic [Rec11, CT10, CR09]. This line of work has the cleanest and strongest theoretical guarantees; [CT10, Rec11] showed that if $|\Omega| \gtrsim dr\mu^2 \log^2 d$ the nuclear norm convex relaxation recovers the exact underlying low rank matrix. The solution can be computed via the solving a convex program in polynomial time. However the primary disadvantage of nuclear norm methods is their computational and memory requirements. The fastest known algorithms have running time $O(d^3)$ and require $O(d^2)$ memory, which are both prohibitive for moderate to large values of $d$. These concerns led to the development of the low-rank factorization paradigm of [BM03]; Burer and Monteiro proposed factorizing the optimization variable $\widehat{M} = XX^T$, and optimizing over $X \in \mathbb{R}^{d \times r}$ instead of $\widehat{M} \in \mathbb{R}^{d \times d}$. This approach only requires $O(dr)$ memory, and a single gradient iteration takes time $O(r|\Omega|)$, so has much lower memory requirement and computational complexity than the nuclear norm relaxation. On the other hand, the factorization causes the optimization problem to be non-convex in $X$, which leads to theoretical difficulties in analyzing algorithms. Under incoherence and sufficient sample size assumptions, [KMO10] showed that well-initialized gradient descent recovers $M$. Similarly, [HW14, Har14, JNS13] showed that well-initialized alternating least squares or block coordinate descent converges to $M$, and [CW15] showed that well-initialized gradient descent converges to $M$. [SL15, ZWL15] provided a more unified analysis by showing that with careful initialization many algorithms, including gradient descent and alternating least squres, succeed. [SL15] accomplished this by showing an analog of strong convexity in the neighborhood of the solution $M$.

**Non-convex Optimization.** Recently, a line of work analyzes non-convex optimization by separating the problem into two aspects: the geometric aspect which shows the function has no spurious local minimum and the algorithmic aspect which designs efficient algorithms can converge to local minimum that satisfy first and (relaxed versions) of second order necessary conditions.

Our result is the first that explains the geometry of the matrix completion objective. Similar geometric results are only known for a few problems: phase retrieval/synchronization, orthogonal tensor decomposition, dictionary learning [GHJY15, SQW15, BBV16]. The matrix completion objective requires different tools due to the sampling of the observed entries, as well as carefully managing the regularizer to restrict the geometry. Parallel to our work Bhojanapalli et al.[BNS16] showed similar results for matrix sensing, which is closely related to matrix completion. Loh and Wainwright [LW15] showed that for many statistical settings that involve missing/noisy data and non-convex regularizers, any stationary point of the non-convex objective is close to global optima; furthermore, there is a unique stationary point that is the global minimum under stronger assumptions [LW14].

On the algorithmic side, it is known that second order algorithms like cubic regularization [NP06] and trust-region [SQW15] algorithms converge to local minima that approximately satisfy first and second order conditions. Gradient descent is also known to converge to local minima [LSJR16] from a random starting point. Stochastic gradient descent can converge to a local minimum in polynomial time from any starting point [Pem90, GHJY15]. All of these results can be applied to our setting, implying various heuristics used in practice are guaranteed to solve matrix completion.

## 2 Preliminaries

**Notations:** For $\Omega \subset [d] \times [d]$, let $P_\Omega$ be the linear operator that maps a matrix $A$ to $P_\Omega(A)$, where $P_\Omega(A)$ has the same values as $A$ on $\Omega$, and 0 outside of $\Omega$. We will use the following matrix norms: $\|\cdot\|_F$ the frobenius norm, $\|\cdot\|$ spectral norm, $|A|_\infty$ elementwise infinity norm, and $|A|_{p \to q} = \max_{\|x\|_p = 1} \|A\|_q$. We use the shorthand $\|A\|_\Omega = \|P_\Omega A\|_F$. The trace inner product of two matrices is $\langle A, B \rangle = \operatorname{tr}(A^\top B)$, and $\sigma_{\min}(X)$, $\sigma_{\max}(X)$ are the smallest and largest singular values of $X$. We also use $X_i$ to denote the $i$-th row of a matrix $X$.

## 2.1 Necessary conditions for Optimality

Given an objective function $f(x) : \mathbb{R}^n \to \mathbb{R}$, we use $\nabla f(x)$ to denote the gradient of the function, and $\nabla^2 f(x)$ to denote the Hessian of the function ($\nabla^2 f(x)$ is an $n \times n$ matrix where $[\nabla^2 f(x)]_{i,j} = \frac{\partial^2}{\partial x_i \partial x_j} f(x)$). It is well known that local minima of the function $f(x)$ must satisfy some necessary conditions:

**Definition 2.1.** *A point $x$ satisfies the first order necessary condition for optimality (later abbreviated as first order optimality condition) if $\nabla f(x) = 0$. A point $x$ satisfies the second order necessary condition for optimality (later abbreviated as second order optimality condition)if $\nabla^2 f(x) \succeq 0$.*

These conditions are necessary for a local minimum because otherwise it is easy to find a direction where the function value decreases. We will also consider a relaxed second order necessary condition, where we only require the smallest eigenvalue of the Hessian $\nabla^2 f(x)$ to be not very negative:

**Definition 2.2.** *For $\tau \geqslant 0$, a point $x$ satisfies the $\tau$-relaxed second order optimality condition, if $\nabla^2 f(x) \succeq -\tau \cdot I$.*

This relaxation to the second order condition makes the conditions more robust, and allows for efficient algorithms.

**Theorem 2.3.** *[NP06, SQW15, GHJY15] If every point $x$ that satisfies first order and $\tau$-relaxed second order necessary condition is a global minimum, then many optimization algorithms (cubic regularization, trust-region, stochastic gradient descent) can find the global minimum up to $\varepsilon$ error in function value in time* $\mathrm{poly}(1/\varepsilon, 1/\tau, d)$.

# 3 Proof Strategy: "simple" proofs are more generalizable

In this section, we demonstrate the key ideas behind our analysis using the rank $r = 1$ case. In particular, we first give a "simple" proof for the fully observed case. Then we show this simple proof can be easily generalized to the random observation case. We believe that this proof strategy is applicable to other statistical problems involving partial/noisy observations. The proof sketches in this section are only meant to be illustrative and may not be fully rigorous in various places. We refer the readers to Section 4 and Section A for the complete proofs.

In the rank $r = 1$ case, we assume $M = zz^{\top}$, where $\|z\| = 1$, and $\|z\|_{\infty} \leqslant \frac{\mu}{\sqrt{d}}$. Let $\varepsilon \ll 1$ be the target accuracy that we aim to achieve in this section and let $p = \mathrm{poly}(\mu, \log d)/(d\varepsilon)$.

For simplicity, we focus on the following domain $\mathcal{B}$ of incoherent vectors where the regularizer $R(x)$ vanishes,

$$\mathcal{B} = \left\{ x : \|x\|_{\infty} < \frac{2\mu}{\sqrt{d}} \right\}. \tag{3.1}$$

Inside this domain $\mathcal{B}$, we can restrict our attention to the objective function without the regularizer, defined as,

$$\tilde{g}(x) = \frac{1}{2} \cdot \|P_{\Omega}(M - xx^{\top})\|_F^2. \tag{3.2}$$

The global minima of $\tilde{g}(\cdot)$ are $z$ and $-z$ with function value 0. Our goal of this section is to (informally) prove that all the local minima of $\tilde{g}(\cdot)$ are $O(\sqrt{\varepsilon})$-close to $\pm z$. In later section we will formally prove that the only local minima are $\pm z$.

**Lemma 3.1** (Partial observation case, informally stated)**.** *Under the setting of this section, in the domain $\mathcal{B}$, all local mimina of the function $\tilde{g}(\cdot)$ are $O(\sqrt{\varepsilon})$-close to $\pm z$.*

It turns out to be insightful to consider the full observation case when $\Omega = [d] \times [d]$. The corresponding objective is

$$g(x) = \frac{1}{2} \cdot \|M - xx^{\top}\|_F^2. \tag{3.3}$$

Observe that $\tilde{g}(x)$ is a sampled version of the $g(x)$, and therefore we expect that they share the same geometric properties. In particular, if $g(x)$ does not have spurious local minima then neither does $\tilde{g}(x)$.

**Lemma 3.2** (Full observation case, informally stated). *Under the setting of this section, in the domain $\mathcal{B}$, the function $g(\cdot)$ has only two local minima $\{\pm z\}$.*

Before introducing the "simple" proof, let us first look at a delicate proof that does not generalize well.

**Difficult to Generalize Proof of Lemma 3.2.** We compute the gradient and Hessian of $g(x)$, $\nabla g(x) = Mx - \|x\|^2 x$, $\nabla^2 g(x) = 2xx^\top - M + \|x\|^2 \cdot I$. Therefore, a critical point $x$ satisfies $\nabla g(x) = Mx - \|x\|^2 x = 0$, and thus it must be an eigenvector of $M$ and $\|x\|^2$ is the corresponding eigenvalue. Next, we prove that the hessian is only positive definite at the top eigenvector. Let $x$ be an eigenvector with eigenvalue $\lambda = \|x\|^2$, and $\lambda$ is strictly less than the top eigenvalue $\lambda^*$. Let $z$ be the top eigenvector. We have that $\langle z, \nabla^2 g(x)z \rangle = -\langle z, Mz \rangle + \|x\|^2 = -\lambda^* + \lambda < 0$, which shows that $x$ is not a local minimum. Thus only $z$ can be a local minimizer, and it is easily verified that $\nabla^2 g(z)$ is indeed positive definite.

The difficulty of generalizing the proof above to the partial observation case is that it uses the *properties of eigenvectors* heavily. Suppose we want to imitate the proof above for the partial observation case, the first difficulty is how to solve the equation $\tilde{g}(x) = P_\Omega(M - xx^\top)x = 0$. Moreover, even if we could have a reasonable approximation for the critical points (the solution of $\nabla \tilde{g}(x) = 0$), it would be difficult to examine the Hessian of these critical points without having the orthogonality of the eigenvectors.

**"Simple" and Generalizable proof.** The lessons from the subsection above suggest us find an alternative proof for the full observation case which is generalizable. The alternative proof will be simple in the sense that it doesn't use the notion of eigenvectors and eigenvalues. Concretely, the key observation behind most of the analysis in this paper is the following,

*Proofs that consist of inequalities that are linear in $\mathbf{1}_\Omega$ are often easily generalizable to partial observation case.*

Here statements that are linear in $\mathbf{1}_\Omega$ mean the statements of the form $\sum_{ij} \mathbf{1}_{(i,j) \in \Omega} T_{ij} \leqslant a$. We will call these kinds of proofs "simple" proofs in this section. Roughly speaking, the observation follows from the law of large numbers — Suppose $T_{ij}, (i, j) \in [d] \times [d]$ is a sequence of bounded real numbers, then the sampled sum $\sum_{(i,j) \in \Omega} T_{ij} = \sum_{i,j} \mathbf{1}_{(i,j) \in \Omega} T_{ij}$ is an accurate estimate of the sum $p \sum_{i,j} T_{ij}$, when the sampling probability $p$ is relatively large. Then, the mathematical implications of $p \sum T_{ij} \leqslant a$ are expected to be similar to the implications of $\sum_{(i,j) \in \Omega} T_{ij} \leqslant a$, up to some small error introduced by the approximation. To make this concrete, we give below informal proofs for Lemma 3.2 and Lemma 3.1 that only consists of statements that are linear in $\mathbf{1}_\Omega$. Readers will see that due to the linearity, the proof for the partial observation case (shown on the right column) is a direct generalization of the proof for the full observation case (shown on the left column) via concentration inequalities (which will be discussed more at the end of the section).

| A "simple" proof for Lemma 3.2. | Generalization to Lemma 3.1. |
|---|---|
| **Claim 1f.** *Suppose $x \in \mathcal{B}$ satisfies $\nabla g(x) = 0$, then $\langle x, z \rangle^2 = \|x\|^4$.* | **Claim 1p.** *Suppose $x \in \mathcal{B}$ satisfies $\nabla \tilde{g}(x) = 0$, then $\langle x, z \rangle^2 = \|x\|^4 - \varepsilon$.* |
| *Proof.* We have, | *Proof.* Imitating the proof on the left, we have |

$$\nabla g(x) = (zz^\top - xx^\top)x = 0 \qquad\qquad \nabla \tilde{g}(x) = P_\Omega(zz^\top - xx^\top)x = 0$$
$$\Rightarrow \langle x, \nabla g(x) \rangle = \langle x, (zz^\top - xx^\top)x \rangle = 0 \qquad \Rightarrow \langle x, \nabla \tilde{g}(x) \rangle = \langle x, P_\Omega(zz^\top - xx^\top)x \rangle = 0$$
$$(3.4) \qquad\qquad\qquad\qquad\qquad (3.5)$$
$$\Rightarrow \langle x, z \rangle^2 = \|x\|^4 \qquad\qquad\qquad \Rightarrow \langle x, z \rangle^2 \geqslant \|x\|^4 - \varepsilon$$

| Intuitively, this proof says that the norm of a critical point $x$ is controlled by its correlation with $z$. | The last step uses the fact that equation (3.4) and (3.5) are approximately equal up to scaling factor $p$ for any $x \in \mathcal{B}$, since (3.5) is a sampled version of (3.4). $\qquad\square$ |

**Claim 2f.** *If $x \in \mathcal{B}$ has positive Hessian $\nabla^2 g(x) \succeq 0$, then $\|x\|^2 \geqslant 1/3$.*

*Proof.* By the assumption on $x$, we have that $\langle z, \nabla^2 g(x)z \rangle \geqslant 0$. Calculating the quadratic form of the Hessian (see Proposition 4.1 for details),

$$\langle z, \nabla^2 g(x)z \rangle$$
$$= \|zx^\top + xz^\top\|^2$$
$$- 2z^\top(zz^\top - xx^\top)z \geqslant 0 \qquad (3.6)$$
$$\Rightarrow \|x\|^2 + 2\langle z, x \rangle^2 \geqslant 1$$
$$\Rightarrow \|x\|^2 \geqslant 1/3 \qquad (\text{since } \langle z, x \rangle^2 \leqslant \|x\|^2)$$

$\square$

**Claim 2p.** *If $x \in \mathcal{B}$ has positive Hessian $\nabla^2 \tilde{g}(x) \succeq 0$, then $\|x\|^2 \geqslant 1/3 - \varepsilon$.*

*Proof.* Imitating the proof on the left, calculating the quadratic form over the Hessian at $z$ (see Proposition 4.1) , we have

$$\langle z, \nabla^2 \tilde{g}(x)z \rangle$$
$$= \|P_\Omega(zx^\top + xz^\top)\|^2$$
$$- 2z^\top P_\Omega(zz^\top - xx^\top)z \geqslant 0 \qquad (3.7)$$
$$\Rightarrow \cdots\cdots \qquad (\text{same step as the left})$$
$$\Rightarrow \|x\|^2 \geqslant 1/3 - \varepsilon$$

Here we use the fact that $\langle z, \nabla^2 \tilde{g}(x)z \rangle \approx p\langle z, \nabla^2 g(x)z \rangle$ for any $x \in \mathcal{B}$. $\square$

With these two claims, we are ready to prove Lemma 3.2 and 3.1 by using another step that is linear in $\mathbf{1}_\Omega$.

*Proof of Lemma 3.2.* By Claim 1f and 2f, we have $x$ satisfies $\langle x, z \rangle^2 \geqslant \|x\|^4 \geqslant 1/9$. Moreover, we have that $\nabla g(x) = 0$ implies

$$\langle z, \nabla g(x) \rangle = \langle z, (zz^\top - xx^\top)x \rangle = 0 \qquad (3.8)$$
$$\Rightarrow \langle x, z \rangle(1 - \|x\|^2) = 0$$
$$\Rightarrow \|x\|^2 = 1 \qquad (\text{by } \langle x, z \rangle^2 \geqslant 1/9)$$

Then by Claim 1f again we obtain $\langle x, z \rangle^2 = 1$, and therefore $x = \pm z$.

$\square$

*Proof of Lemma 3.1.* By Claim 1p and 2p, we have $x$ satisfies $\langle x, z \rangle^2 \geqslant \|x\|^4 \geqslant 1/9 - O(\varepsilon)$. Moreover, we have that $\nabla \tilde{g}(x) = 0$ implies

$$\langle z, \nabla \tilde{g}(x) \rangle = \langle z, P_\Omega(zz^\top - xx^\top)x \rangle = 0 \qquad (3.9)$$
$$\Rightarrow \cdots\cdots \qquad (\text{same step as the left})$$
$$\Rightarrow \|x\|^2 = 1 \pm O(\varepsilon) \qquad (\text{same step as the left})$$

Since (3.9) is the sampled version of equation (3.8), we expect they lead to the same conclusion up to some approximation. Then by Claim 1p again we obtain $\langle x, z \rangle^2 = 1 \pm O(\varepsilon)$, and therefore $x$ is $O(\sqrt{\varepsilon})$-close to either of $\pm z$. $\square$

**Subtleties regarding uniform convergence.** In the proof sketches above, our key idea is to use concentration inequalities to link the full observation objective $g(x)$ with the partial observation counterpart. However, we require a uniform convergence result. For example, we need a statement like "w.h.p over the choice of $\Omega$, equation (3.4) and (3.5) are similar to each other up to scaling". This type of statement is often only true for $x$ inside the incoherent ball $\mathcal{B}$. The fix to this is the regularizer. For non-incoherent $x$, we will use a different argument that uses the property of the regularizer. This is besides the main proof strategy of this section and will be discussed in subsequent sections.

## 4 Warm-up: Rank-1 Case

In this section, using the general proof strategy described in previous section, we provide a formal proof for the rank-1 case. In subsection 4.1, we formally work out the proof sketches of Section 3 inside the incoherent ball. The rest of the proofs is deferred to supplementary material.

In the rank-1 case, the objective function simplifies to,

$$f(x) = \frac{1}{2}\|P_\Omega(M - xx^\top)\|_F^2 + \lambda R(x). \qquad (4.1)$$

Here we use the the regularization $R(x)$

$$R(x) = \sum_{i=1}^{d} h(x_i), \text{ and } h(t) = (|t| - \alpha)^4 \, \mathbb{I}_{t \geqslant \alpha} \ .$$

The parameters $\lambda$ and $\alpha$ will be chosen later as in Theorem 4.2. We will choose $\alpha > 10\mu/\sqrt{d}$ so that $R(x) = 0$ for incoherent $x$, and thus it only penalizes coherent $x$. Moreover, we note $R(x)$ has Lipschitz second order derivative. [3]

We first state the optimality conditions, whose proof is deferred to Appendix A.

**Proposition 4.1.** *The first order optimality condition of objective* (4.1) *is,*

$$2P_\Omega(M - xx^\top)x = \lambda \nabla R(x) \,, \tag{4.2}$$

*and the second order optimality condition requires:*

$$\forall v \in \mathbb{R}^d, \ \|P_\Omega(vx^\top + xv^\top)\|_F^2 + \lambda v^\top \nabla^2 R(x)v \geqslant 2v^\top P_\Omega(M - xx^\top)v \,. \tag{4.3}$$

*Moreover, The $\tau$-relaxed second order optimality condition requires*

$$\forall v \in \mathbb{R}^d, \ \|P_\Omega(vx^\top + xv^\top)\|_F^2 + \lambda v^\top \nabla^2 R(x)v \geqslant 2v^\top P_\Omega(M - xx^\top)v - \tau \|v\|^2 \,. \tag{4.4}$$

We give the precise version of Theorem 1.1 for the rank-1 case.

**Theorem 4.2.** *For $p \geqslant \frac{c\mu^6 \log^{1.5} d}{d}$ where $c$ is a large enough absolute constant, set $\alpha = 10\mu\sqrt{1/d}$ and $\lambda \geqslant \mu^2 p/\alpha^2$. Then, with high probability over the randomness of $\Omega$, the only points in $\mathbb{R}^d$ that satisfy both first and second order optimality conditions (or $\tau$-relaxed optimality conditions with $\tau < 0.1p$) are $z$ and $-z$.*

In the rest of this section, we will first prove that when $x$ is constrained to be incoherent (and hence the regularizer is 0 and concentration is straightforward) and satisfies the optimality conditions, then $x$ has to be $z$ or $-z$. Then we go on to explain how the regularizer helps us to change the geometry of those points that are far away from $z$ so that we can rule out them from being local minimum. For simplicity, we will focus on the part that shows a local minimum $x$ must be close enough to $z$.

**Lemma 4.3.** *In the setting of Theorem 4.2, suppose $x$ satisfies the first-order and second-order optimality condition* (4.2) *and* (4.3)*. Then when $p$ is defined as in Theorem 4.2,*

$$\left\| xx^\top - zz^\top \right\|_F^2 \leqslant O(\varepsilon) \,.$$

*where $\varepsilon = \mu^3 (pd)^{-1/2}$.*

This turns out to be the main challenge. Once we proved $x$ is close, we can apply the result of Sun and Luo [SL15] (see Lemma C.1), and obtain Theorem 4.2.

## 4.1 Handling incoherent $x$

To demonstrate the key idea, in this section we restrict our attention to the subset of $\mathbb{R}^d$ which contains incoherent $x$ with $\ell_2$ norm bounded by 1, that is, we consider,

$$\mathcal{B} = \left\{ x : \|x\|_\infty \leqslant \frac{2\mu}{\sqrt{d}}, \|x\| \leqslant 1 \right\} \,. \tag{4.5}$$

Note that the desired solution $z$ is in $\mathcal{B}$, and the regularization $R(x)$ vanishes inside $\mathcal{B}$.

The following lemmas assume $x$ satisfies the first and second order optimality conditions, and deduce a sequence of properties that $x$ must satisfy.

**Lemma 4.4.** *Under the setting of Theorem 4.2 , with high probability over the choice of $\Omega$, for any $x \in \mathcal{B}$ that satisfies second-order optimality condition* (4.3) *we have,*

$$\|x\|^2 \geqslant 1/4.$$

*The same is true if $x \in \mathcal{B}$ only satisfies $\tau$-relaxed second order optimality condition for $\tau \leqslant 0.1p$.*

*Proof.* We plug in $v = z$ in the second-order optimality condition (4.3), and obtain that

$$\left\| P_\Omega(zx^\top + xz^\top) \right\|_F^2 \geqslant 2z^\top P_\Omega(M - xx^\top)z \,. \tag{4.6}$$

Intuitively, when restricted to $\Omega$, the squared Frobenius on the LHS and the quadratic form on the RHS should both be approximately a $p$ fraction of the unrestricted case. In fact, both LHS and RHS can be written as the sum of terms of the form $\langle P_\Omega(uv^T), P_\Omega(st^T)\rangle$, because

$$\left\|P_\Omega(zx^\top + xz^\top)\right\|_F^2 = 2\langle P_\Omega(zx^T), P_\Omega(zx^T)\rangle + 2\langle P_\Omega(zx^T), P_\Omega(xz^T)\rangle$$
$$2z^\top P_\Omega(M - xx^\top)z = 2\langle P_\Omega(zz^T), P_\Omega(zz^T)\rangle - 2\langle P_\Omega(xx^T), P_\Omega(zz^T)\rangle.$$

Therefore we can use concentration inequalities (Theorem D.1), and simplify the equation

$$\text{LHS of (4.6)} = p\left\|zx^\top + xz^\top\right\|_F^2 \pm O(\sqrt{pd\|x\|_\infty^2\|z\|_\infty^2\|x\|^2\|z\|^2})$$
$$= 2p\|x\|^2\|z\|^2 + 2p\langle x, z\rangle^2 \pm O(p\varepsilon), \qquad\qquad \text{(Since } x, z \in \mathcal{B})$$

where $\varepsilon = O(\mu^2\sqrt{\frac{\log d}{pd}})$. Similarly, by Theorem D.1 again, we have

$$\text{RHS of (4.6)} = 2\left(\langle P_\Omega(zz^\top), P_\Omega(zz^\top)\rangle - \langle P_\Omega(xx^\top), P_\Omega(zz^\top)\rangle\right) \qquad \text{(Since } M = zz^\top)$$
$$= 2p\|z\|^4 - 2p\langle x, z\rangle^2 \pm O(p\varepsilon) \qquad\qquad \text{(by Theorem D.1 and } x, z \in \mathcal{B})$$

(Note that even we use the $\tau$-relaxed second order optimality condition, the RHS only becomes $1.99p\|z\|^4 - 2p\langle x, z\rangle^2 \pm O(p\varepsilon)$ which does not effect the later proofs.)

Therefore plugging in estimates above back into equation (4.6), we have that

$$2p\|x\|^2\|z\|^2 + 2p\langle x, z\rangle^2 \pm O(p\varepsilon) \geqslant 2\|z\|^4 - 2\langle x, z\rangle^2 \pm O(p\varepsilon),$$

which implies that $6p\|x\|^2\|z\|^2 \geqslant 2p\|x\|^2\|z\|^2 + 4p\langle x, z\rangle^2 \geqslant 2p\|z\|^4 - O(p\varepsilon)$. Using $\|z\|^2 = 1$, and $\varepsilon$ being sufficiently small, we complete the proof. $\qquad\square$

Next we use first order optimality condition to pin down another property of $x$ – it has to be close to $z$ after scaling. Note that this doesn't mean directly that $x$ has to be close to $z$ since $x = 0$ also satisfies first order optimality condition (and therefore the conclusion (4.7) below).

**Lemma 4.5.** *With high probability over the randomness of $\Omega$, for any $x \in \mathcal{B}$ that satisfies first-order optimality condition* (4.2)*, we have that $x$ also satisfies*

$$\left\|\langle z, x\rangle z - \|x\|^2 x\right\| \leqslant O(\varepsilon). \qquad\qquad (4.7)$$

*where $\varepsilon = \tilde{O}(\mu^3(pd)^{-1/2})$.*

Finally we combine the two optimality conditions and show equation (4.7) implies $xx^T$ must be close to $zz^T$.

**Lemma 4.6.** *Suppose vector $x$ satisfies that $\|x\|^2 \geqslant 1/4$, and that $\left\|\langle z, x\rangle z - \|x\|^2 x\right\| \leqslant \delta$. Then for $\delta \in (0, 0.1)$,*

$$\left\|xx^\top - zz^\top\right\|_F^2 \leqslant O(\delta).$$

# 5 Conclusions

Although the matrix completion objective is non-convex, we showed the objective function has very nice properties that ensures the local minima are also global. This property gives guarantees for many basic optimization algorithms. An important open problem is the robustness of this property under different model assumptions: Can we extend the result to handle asymmetric matrix completion? Is it possible to add weights to different entries (similar to the settings studied in [LLR16])? Can we replace the objective function with a different distance measure rather than Frobenius norm (which is related to works on 1-bit matrix sensing [DPvdBW14])? We hope this framework of analyzing the geometry of objective function can be applied to other problems.

## Footnotes

[1]In this paper, we focus on the symmetric case when the true $M$ has a symmetric decomposition $M = ZZ^T$. Some of previous papers work on the asymmetric case when $M = ZW^T$, which is harder than the symmetric case.

[2]The entries $(i, j)$ and $(j, i)$ are the same. With probability $p$ we observe both entries and otherwise we observe neither.

[3]This is the main reason for us to choose 4-th power instead of 2-nd power.

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
