[Supplementary Material]

# Matrix Completion has No Spurious Local Minimum

Rong Ge[*]        Jason D. Lee[†]        Tengyu Ma[‡]

October 25, 2016

**Abstract**

Matrix completion is a basic machine learning problem that has wide applications, especially in collaborative filtering and recommender systems. Simple non-convex optimization algorithms are popular and effective in practice. Despite recent progress in proving various non-convex algorithms converge from a good initial point, it remains unclear why random or arbitrary initialization suffices in practice. We prove that the commonly used non-convex objective function for *positive semidefinite* matrix completion has no spurious local minima – all local minima must also be global. Therefore, many popular optimization algorithms such as (stochastic) gradient descent can provably solve positive semidefinite matrix completion with *arbitrary* initialization in polynomial time. The result can be generalized to the setting when the observed entries contain noise. We believe that our main proof strategy can be useful for understanding geometric properties of other statistical problems involving partial or noisy observations.

## 1 Introduction

Matrix completion is the problem of recovering a low rank matrix from partially observed entries. It has been widely used in collaborative filtering and recommender systems [Kor09, RS05], dimension reduction [CLMW11] and multi-class learning [AFSU07]. There has been extensive work on designing efficient algorithms for matrix completion with guarantees. One earlier line of results (see [Rec11, CT10, CR09] and the references therein) rely on convex relaxations. These algorithms achieve strong statistical guarantees, but are quite computationally expensive in practice.

More recently, there has been growing interest in analyzing non-convex algorithms for matrix completion [KMO10, JNS13, Har14, HW14, SL15, ZWL15, CW15]. Let $M \in \mathbb{R}^{d \times d}$ be the target matrix with rank $r \ll d$ that we aim to recover, and let $\Omega = \{(i, j) : M_{i,j} \text{ is observed}\}$ be the set of observed entries. These methods are instantiations of optimization algorithms applied to the objective[1],

$$f(X) = \frac{1}{2} \sum_{(i,j) \in \Omega} \left[ M_{i,j} - (XX^\top)_{i,j} \right]^2, \tag{1.1}$$

These algorithms are much faster than the convex relaxation algorithms, which is crucial for their empirical success in large-scale collaborative filtering applications [Kor09].

All of the theoretical analysis for the nonconvex procedures require careful initialization schemes: the initial point should already be close to optimum. In fact, Sun and Luo [SL15] showed that after this initialization the problem is effectively strongly-convex, hence many different optimization procedures can be analyzed by standard techniques from convex optimization.

However, in practice people typically use a random initialization, which still leads to robust and fast convergence. Why can these practical algorithms find the optimal solution in spite of the non-convexity? In this work we investigate this question and show that the matrix completion objective has *no spurious* local minima. More precisely, we show

---

[*]Duke University, rongge@cs.duke.edu.

[†]University of Southern California, jasonlee@marshall.usc.edu.

[‡]Princeton University, tengyu@cs.princeton.edu. Supported in part by Simons Award in Theoretical Computer Science and IBM PhD Fellowship.

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

*where $\varepsilon = \tilde{O}(\mu^3(pd)^{-1/2})$.*

*Proof.* Note that since $x \in \mathcal{B}$, we have $R(x) = 0$. Therefore first-order optimality condition says that

$$P_\Omega(M - xx^\top)x = P_\Omega(zz^\top)x - P_\Omega(xx^\top)x = 0. \tag{4.8}$$

Again, intuitively we hope $P_\Omega(zz^T) \approx pzz^T$ and $P_\Omega(xx^T)x \approx p\|x\|^2 x$. These are made precise by the concentration inequalities Lemma D.4 and Theorem D.2 respectively.

By Theorem D.2, we have that with high probability over the choice of $\Omega$, for every $x \in \mathcal{B}$,

$$\|P_\Omega(xx^\top)x - pxx^\top x\|_F \leqslant p\varepsilon\|x\|^3 \leqslant p\varepsilon \tag{4.9}$$

where $\varepsilon = \tilde{O}(\mu^3(pd)^{-1/2})$. Similarly, by Lemma D.4, we have that for with high probability over the choice of $\Omega$,

$$\left\|P_\Omega(zz^\top) - pzz^\top\right\| \leqslant \varepsilon p\,.$$

for $\varepsilon = \tilde{O}(\mu^2(pd)^{-1/2})$. Therefore for every $x$,

$$\left\|P_\Omega(zz^\top)x - pzz^\top x\right\| \leqslant \varepsilon p\|x\| \leqslant \varepsilon p\,. \tag{4.10}$$

Plugging in estimates (4.10) and (4.9) into equation (4.8), we complete the proof. $\qquad\square$

Finally we combine the two optimality conditions and show equation (4.7) implies $xx^T$ must be close to $zz^T$.

**Lemma 4.6.** *Suppose vector $x$ satisfies that $\|x\|^2 \geqslant 1/4$, and that $\left\|\langle z,x\rangle z - \|x\|^2 x\right\| \leqslant \delta$. Then for $\delta \in (0, 0.1)$,*

$$\left\|xx^\top - zz^\top\right\|_F^2 \leqslant O(\delta)\,.$$

*Proof.* We write $z = ux + v$ where $u \in \mathbb{R}$ and $v$ is a vector orthogonal to $x$. Now we know $\langle z, x\rangle z = u^2\|x\|^2 x + u\|x\|^2 v$, therefore

$$\delta \geqslant \left\|\langle z,x\rangle z - \|x\|^2 x\right\| = \|x\|^2\sqrt{u^2\|v\|^2 + (1-u^2)^2}.$$

In particular, we know $|1 - u^2| \leqslant 4\delta$ and $u\|v\| \leqslant 4\delta$. This means $|u| \in 1 \pm 3\delta$ and $\|v\| \leqslant 8\delta$. Now we expand $xx^T - zz^T$:

$$xx^T - zz^T = (1 - u^2)xx^T + uxv^T + uvx^T + vv^T$$

It is clear that all the terms have norm bounded by $O(\delta)$, therefore $\left\|xx^\top - zz^\top\right\|_F^2 \leqslant O(\delta)$. $\qquad\square$

## 4.2 Extension to general $x$

We have shown when $x$ is incoherent and satisfies first and second order optimality conditions, then it must be close to $z$ or $-z$. Now we need to consider more general cases when $x$ may have some very large coordinates. Here the main intuition is that the first order optimality condition with a proper regularizer is enough to guarantee that $x$ cannot have a entry that is too much bigger than $\mu/\sqrt{d}$.

**Lemma 4.7.** *With high probability over the choice of $\Omega$, for any $x$ that satisfies first-order order optimality condition (4.2), we have*

$$\|x\|_\infty \leqslant 4\max\left\{\alpha, \mu\sqrt{p/\lambda}\right\}\,. \tag{4.11}$$

Here we recall that $\alpha$ was chosen to be $10\mu/\sqrt{d}$ and $\lambda$ is chosen to be large so that the $\alpha$ dominates the second term $\mu\sqrt{p/\lambda}$ in the setting of Theorem 4.2.

*Proof of Lemma 4.7.* Suppose $i^\star = \max_j |x_j|$. Without loss of generality, suppose $x_{i^\star} \geqslant 0$. Suppose $i^\star$-th row of $\Omega$ consists of entries with index $[i] \times S_{i^\star}$. If $|x_{i^\star}| \leqslant 2\alpha$, we are done. Therefore in the rest of the proof we assume $|x_{i^\star}| > 2\alpha$. Note that when $p \geqslant c(\log d)/d$ for sufficiently large constant $c$, with high probability over the choice of $\Omega$, we have $|S_{i^\star}| \leqslant 2pd$. In the rest of argument we are working with such an $\Omega$ with $|S_{i^\star}| \leqslant 2pd$.

We will compare the $i^\star$-th coordinate of LHS and RHS of first-order optimality condition (4.2). For preparation, we have

$$\left|(P_\Omega(M)x)_{i^\star}\right| = \left|\left(P_\Omega(zz^\top)x\right)_{i^\star}\right| = \left|\sum_{j \in S_{i^\star}} z_{i^\star}z_j x_j\right|$$

$$\leqslant |x_{i^\star}|\sum_{j \in S_{i^\star}} |z_{i^\star}z_j| \leqslant |x_{i^\star}| \cdot \mu^2/d \cdot |S_{i^\star}| \leqslant 2|x_{i^\star}|p\mu^2 \tag{4.12}$$

objective $f(x) = \|P_\Omega(M) - P_\Omega(xx^\top)\|_F^2 + \lambda R(x)$

$\mathbb{R}^d$

Lemma 4.7:
$\nabla f(x) \neq 0$

$z$

Lemma 4.10:
$\nabla f(x) \neq 0$

Lemma 4.8
$\nabla^2 f(x) \not\succeq 0$

$-z$

$\left\{ x : \|x\|_\infty \leq \dfrac{4\mu}{\sqrt{d}} \right\}$   ● local (and global) min

$\left\{ x : \|x\|^2 \leq \dfrac{1}{16} \right\}$

Figure 1: Partition of $\mathbb{R}^d$ into regions where our Lemmas apply. For example, Lemma 3.8 rules out the possibility that a point $x$ in the green region is local minimum. Here, The green region is the intersection of $\ell_\infty$ norm ball and $\ell_2$ norm ball. Both the white region and yellow region have non-zero gradient but for different reasons.

where the last step we used the fact that $|S_{i^\star}| \leq 2pd$. Moreover, we have that

$$(P_\Omega(xx^\top)x)_{i^\star} = \sum_{j \in S_{i^\star}} x_{i^\star} x_j^2 \geq 0,$$

and that

$$(\lambda \nabla R(x))_{i^\star} = 4\lambda(|x_{i^\star}| - \alpha)^3 \operatorname{sign}(x_{i^\star}) \geq \frac{\lambda}{2}|x_{i^\star}|^3 \qquad\qquad \text{(Since } x_{i^\star} \geq 2\alpha\text{)}$$

Now plugging in the bounds above into the $i^\star$-th coordinate of equation (4.2), we obtain

$$4|x_{i^\star}|p\mu^2 \geq 2(P_\Omega(M - xx^\top)x)_{i^\star} \geq (\lambda \nabla R(x))_{i^\star} \geq \frac{\lambda}{2}|x_{i^\star}|^3,$$

which implies that $|x_{i^\star}| \leq 4\sqrt{p\mu^2/\lambda}$. $\qquad\qquad\qquad\qquad\qquad\qquad\qquad\qquad\qquad\qquad\square$

Setting $\lambda \geq \mu^2 p/\alpha^2$ and $\alpha = 10\mu\sqrt{1/d}$, Lemma 4.7 ensures that any $x$ that satisfies first-order optimality condition is the following ball,

$$\mathcal{B}' = \left\{ x \in \mathbb{R}^d : \|x\|_\infty \leq 4\alpha \right\}.$$

Then we would like to continue to use arguments similar to Lemma 4.4 and 4.5. However, things have become more complicated as now we need to consider the contribution of the regularizer.

**Lemma 4.8** (Extension of Lemma 4.4). *In the setting of Theorem 4.2, with high probability over the choice of $\Omega$, suppose $x \in \mathcal{B}'$ satisfies second-order optimality condition* (4.3) *or $\tau$-relaxed condition for $\tau \leq 0.1p$, we have $\|x\|^2 \geq 1/8$.*

The guarantees and proofs are very similar to Lemma 4.4. The main intuition is that we can restrict our attentions to coordinates whose regularizer is equal to 0. See Section A for details.

We will now deal with first order optimality condition. We first write out the basic extension of Lemma 4.5, which follows from the same proof except we now include the regularizer term.

**Lemma 4.9** (Basic extension of Lemma 4.5)**.** *With high probability over the randomness of $\Omega$, for any $x \in \mathcal{B}'$ that satisfies first-order optimality condition* (4.2)*, we have that $x$ also satisfies*

$$\left\| \langle z, x \rangle z - \|x\|^2 x - \gamma \cdot \nabla R(x) \right\| \leqslant O(\varepsilon) \,. \tag{4.13}$$

*where $\varepsilon = \tilde{O}(\mu^6 (pd)^{-1/2})$ and $\gamma = \lambda/(2p) \geqslant 0$.*

Next we will show that we can remove the regularizer term, the main observation here is nonzero entries $\nabla R(x)$ all have the same sign as the corresponding entries in $x$. See Section A for details.

**Lemma 4.10.** *Suppose $x \in \mathcal{B}'$ satisfies that $\|x\|^2 \geqslant 1/8$, under the same assumption as Lemma 4.9. we have,*

$$\left\| \langle x, z \rangle z - \|x\|^2 x \right\| \leqslant O(\varepsilon)$$

Finally we combine Lemma 4.7, Lemma 4.8, Lemma 4.10 and Lemma 4.6 to prove Lemma 4.3. The argument are also summarized in Figure 1, where we partition $\mathbb{R}^d$ into regions where our lemmas apply.

# 5 Rank-r case

In this section we show how to extend the results in Section 4 to recover matrices of rank $r$. Here we still use the same proof strategy of Section 3. Though for simplicity we only write down the proof for the partial observation case, while the analysis for the full observation case (which was our starting point) can be obtained by substituting $[d] \times [d]$ for $\Omega$ everywhere.

Recall that in this case we assume the original matrix $M = ZZ^T$, where $Z \in \mathbb{R}^{d \times r}$. We also assume Assumption 1. The objective function is very similar to the rank 1 case

$$f(X) = \frac{1}{2} \left\| P_\Omega(M - XX^\top) \right\|_F^2 + \lambda R(X) \,, \tag{5.1}$$

where $R(X) = \sum_{i=1}^d r(\|X_i\|)$. Recall that $r(t) = (|t| - \alpha)^4 \mathbb{I}_{t \geqslant \alpha}$. Here $\alpha$ and $\lambda$ are again parameters that we will determined later.

Without loss of generality, we assume that $\|Z\|_F^2 = r$ in this section. This implies that $\sigma_{\max}(Z) \geqslant 1 \geqslant \sigma_{\min}(Z)$. Now we shall state the first and second order optimality conditions:

**Proposition 5.1.** *If $X$ is a local optimum of objective function* (5.1)*, its first order optimality condition is,*

$$2P_\Omega(M)X = 2P_\Omega(XX^\top)X + \lambda \nabla R(X) \,, \tag{5.2}$$

*and the second order optimality condition is equivalent to*

$$\forall V \in \mathbb{R}^{d \times r}, \ \|P_\Omega(VX^\top + XV^\top)\|_F^2 + \lambda \langle V^\top, \nabla^2 R(X)V \rangle \geqslant 2 \langle P_\Omega(M - XX^\top), VV^\top \rangle \,. \tag{5.3}$$

Note that the regularizer now is more complicated than the one dimensional case, but luckily we still have the following nice property.

**Proposition 5.2.** *We have that $\nabla R(X) = \Gamma X$ where $\Gamma \in \mathbb{R}^{d \times d}$ is a diagonal matrix with $\Gamma_{ii} = \frac{4(\|X_i\| - \alpha)^4}{\|X_i\|} \mathbb{I}_{\|X_i\| \geqslant \alpha}$. As a direct consequence, $\langle (\nabla R(X))_i, X_i \rangle \geqslant 0$ for every $i \in [d]$.*

Now we are ready to state the precise version of Theorem 1.1:

**Theorem 5.3.** *Suppose $p \geqslant C \max\{\mu^6 \kappa^{16} r^4, \mu^4 \kappa^4 r^6\} d^{-1} \log^{1.5} d$ where $C$ is a large enough constant. Let $\alpha = 4\mu\kappa r/\sqrt{d}, \lambda \geqslant \mu^2 r p/\alpha^2$. Then with high probability over the randomness of $\Omega$, any local minimum $X$ of $f(\cdot)$ satisfies that $f(X) = 0$, and in particular, $ZZ^\top = XX^\top$.*

The proof of this Theorem follows from a similar path as Theorem 4.2. We first notice that because of the regularizer, any matrix $X$ that satisfies first order optimality condition must be somewhat incoherent (this is analogues to Lemma 4.7):

**Lemma 5.4.** *Suppose $|S_i| \leqslant 2pd$. Then for any $X$ satisfies 1st order optimality (5.2), we have*

$$\|X\|_{2\to\infty} = \max_i \|X_i\| \leqslant 4\max\left\{\alpha, \mu\sqrt{rp/\lambda}\right\} \tag{5.4}$$

*Proof.* Assume $i^\star = \operatorname{argmax}_i \|X_i\|$. Suppose the $i$th row of $\Omega$ consists of entries with index $[i] \times S_i$. If $\|X_{i^\star}\| \leqslant 2\alpha$, then we are done. Therefore in the rest of the proof we assume $\|X_{i^\star}\| \geqslant 2\alpha$.

We will compare the $i$-th row of LHS and RHS of (5.2). For preparation, we have

$$\left(P_\Omega(M)x\right)_{i^\star} = \left(P_\Omega(ZZ^\top)X\right)_{i^\star} = \left(P_\Omega(ZZ^\top)\right)_{i^\star} X \tag{5.5}$$

Then we have that

$$
\begin{aligned}
\left\|\left(P_\Omega(ZZ^\top)\right)_{i^\star}\right\|_1 &= \sum_{j\in S_{i^\star}} |\langle Z_{i^\star}, Z_j\rangle| \\
&\leqslant \sum_{j\in S_{i^\star}} \|Z_{i^\star}\|\|Z_j\| \leqslant \sum_{j\in S_{i^\star}} \mu^2 r/d |S_1| \qquad &\text{(by incoherence of } Z) \\
&\leqslant 2\mu^2 rp. &\text{(by } |S_{i^\star}| \leqslant 2pd)
\end{aligned}
$$

Therefore we can bound the $\ell_2$ norm of LHS of 1st order optimality condition (5.2) by

$$
\begin{aligned}
\left\|\left(P_\Omega(ZZ^\top)X\right)_{i^\star}\right\| &\leqslant \left\|\left(P_\Omega(ZZ^\top)\right)_{i^\star}\right\|_1 \left\|X^\top\right\|_{1\to 2} \\
&\leqslant 2\mu^2 rp \|X\|_{2\to\infty} \qquad &\text{(by } \|X\|_{2\to\infty} = \left\|X^\top\right\|_{1\to 2}) \\
&= 2\mu^2 rp \|X_{i^\star}\| &(5.6)
\end{aligned}
$$

Next we lowerbound the norm of the RHS of equation (5.2). We have that

$$(P_\Omega(XX^\top)X)_{i^\star} = \sum_{j\in S_{i^\star}} \langle X_{i^\star}, X_j\rangle X_j = X_i \sum_{j\in X_{i^\star}} X_j^\top X_j,$$

which implies that

$$\langle (P_\Omega(XX^\top)X)_{i^\star}, X_{i^\star}\rangle = X_{i^\star}\left(\sum_{j\in X_{i^\star}} X_j^\top X_j\right) X_{i^\star}^\top \geqslant 0. \tag{5.7}$$

Using Proposition 5.2 we obtain that

$$\langle (P_\Omega(XX^\top)X)_{i^\star}, (\nabla R(X))_{i^\star}\rangle = \Gamma_{ii} X_{i^\star}\left(\sum_{j\in X_{i^\star}} X_j^\top X_j\right) X_{i^\star}^\top \geqslant 0. \tag{5.8}$$

It follows that

$$
\begin{aligned}
\left\|(P_\Omega(XX^\top)X)_{i^\star} + (\lambda\nabla R(X))_{i^\star}\right\| &\geqslant \|(\lambda\nabla R(X))_{i^\star}\| \qquad &\text{(by equation (5.8))} \\
&= \frac{4\lambda(\|X_{i^\star}\| - \alpha)^3}{\|X_{i^\star}\|} \cdot \|X_{i^\star}\| &\text{(by Proposition 5.2)} \\
&\geqslant \frac{\lambda}{2}\|X_{i^\star}\|^3 &\text{(by the assumptino } \|X_{i^\star}\| \geqslant 2\alpha)
\end{aligned}
$$

Therefore plugging in equation above and equation (5.6) into 1st order optimality condition (5.2). We obtain that $\|X_{i^\star}\| \leqslant \sqrt{8\mu^2 rp/\lambda}$ which completes the proof. □

Next, we prove a property implied by first order optimality condition, which is similar to Lemma 4.9.

**Lemma 5.5.** *In the setting of Theorem 5.3, with high probability over the choice of $\Omega$, for any $X$ that satisfies 1st order optimality condition* (5.2), *we have*

$$\|X\|_F^2 \leqslant 2r\sigma_{\max}(Z)^2 \,. \tag{5.9}$$

*Moreover, we have*

$$\sigma_{\max}(X) \leqslant 2\sigma_{\max}(Z)r^{1/6} \,. \tag{5.10}$$

*and*

$$\left\| ZZ^T X - XX^T X - \gamma \nabla R(X) \right\|_F \leqslant O(\delta) \tag{5.11}$$

*where $\delta = O(\mu^3 \kappa^3 r^2 \log^{0.75}(d)\sigma_{\max}(Z)^{-3}(dp)^{-1/2})$ and $\gamma = \lambda/(2p) \geqslant 0$.*

*Proof.* If $\|X\|_F \leqslant \sqrt{r\sigma_{\max}(Z)^2}$ we are done. When $\|X\|_F \geqslant \sqrt{r\sigma_{\max}(Z)^2}$, by Lemma 5.4, we have that $\max \|X_i\| \leqslant 4\alpha = 4\mu\kappa r/\sqrt{d}$, and therefore $\max \|X_i\| \leqslant \nu\|X\|_F$ with $\nu = O(\mu\kappa\sqrt{r}/\sigma_{\max}(Z))$. Then by Theorem D.2, we have that

$$\left\| P_\Omega(ZZ^\top)X - pZZ^\top X \right\|_F \leqslant p\delta \,,$$

and

$$\left\| P_\Omega(XX^\top)X - XX^\top X \right\|_F \leqslant p\delta \,,$$

where $\delta = O(\mu^3 \kappa^3 r^2 \log^{0.75}(d)\sigma_{\max}(Z)^{-3}(dp)^{-1/2})$. These two imply equation (5.11). Moreover, we have

$$\begin{aligned}
p \left\| ZZ^\top X \right\|_F = \left\| P_\Omega(ZZ^\top)X \right\|_F \pm p\delta &= \left\| P_\Omega(XX^\top)X + \lambda R(X) \right\|_F \pm p\delta && \text{(by equation (5.2))} \\
&\geqslant \left\| P_\Omega(XX^\top)X \right\|_F \pm p\delta && \text{(by equation (5.8))} \\
&\geqslant p \left\| XX^\top X \right\|_F \pm 2p\delta && (5.12)
\end{aligned}$$

Suppose $X$ has singular value $\sigma_1 \geqslant \ldots \geqslant \sigma_r$. Then we have $\left\| ZZ^\top X \right\|_F^2 \leqslant \|ZZ^\top\|^2 \|X\|_F^2 \leqslant \sigma_{\max}(Z)^4 \|X\|_F^2 = \sigma_{\max}(Z)^4(\sigma_1^2 + \cdots + \sigma_r^2)$. On the other hand, $\left\| XX^\top X \right\|_F^2 = \sigma_1^6 + \cdots + \sigma_r^6$. Therefore, equation (5.12) implies that

$$(1 + O(\delta))\sigma_{\max}(Z)^4 \sum_{i=1}^r \sigma_i^2 \geqslant \sum_{i=1}^r \sigma_i^6$$

Then we have (by Proposition E.1) we complete the proof.

$\square$

Now we look at the second order optimality condition, this condition implies the smallest singular value of $X$ is large (similar to Lemma 4.8). Note that this lemma is also true even if $x$ only satisfies relaxed second order optimality condition with $\tau = 0.01p\sigma_{\min}(Z)$.

**Lemma 5.6.** *In the setting of Theorem 5.3. With high probability over the choice of $\Omega$, suppose $X$ satisfies equation* (5.9), (5.4) *the 2nd order optimality condition* (5.3). *Then,*

$$\sigma_{\min}(X) \geqslant \frac{1}{4}\sigma_{\min}(Z) \tag{5.13}$$

*Proof.* Let $J = \{i : \|X_i\| \leqslant \alpha\}$. Let $v \in \mathbb{R}^r$ such that $\|Xv\| = \sigma_{\min}(X)$. . Let $Z_J$ be the matrix that has the same $i$-th row as $Z$ for every $i \in J$ and 0 elsewhere. Since $Z_J$ has column rank at most $r$, by variational characterization of singular values, we have that for there exists unit vector $z_J \in \text{col-span}(Z_J)$ such that $\|X^\top z_J\| \leqslant \sigma_{\min}(X)$.

We claim that $\sigma_{\min}(Z_J) \geqslant \frac{1}{2}\sigma_{\min}(Z)$. Let $L = [d] - J$. Since for any $i \in L$ it holds that $\|X_i\| \geqslant \alpha$, we have $|L|\alpha^2 \leqslant \|X\|_F^2 \leqslant 2r\sigma_{\max}(Z)^2$ (by equation (5.9)), and it follows that $|L| \leqslant 2r\sigma_{\max}(Z)^2/\alpha^2$. Therefore,

$$\begin{aligned}
\sigma_{\min}(Z_J) \geqslant \sigma_{\min}(Z) - \sigma_{\max}(Z_L) &\geqslant \sigma_{\min}(Z) - \|Z_L\|_F \\
&\geqslant \sigma_{\min}(Z) - \sqrt{|L|r\mu^2/d} \geqslant \sigma_{\min}(Z) - \sqrt{2r^2\sigma_{\max}(Z)^2\mu^2/(\alpha^2 d)} \\
&\geqslant \frac{1}{2}\sigma_{\min}(Z) \,. && (\text{by } \alpha \geqslant \tfrac{r\kappa\mu}{\sqrt{d}})
\end{aligned}$$

Since $z_J \in$ col-span$(Z_J)$ is a unit vector, we have that $z_J$ can be written as $z_J = Z_J\beta$ where $\|\beta\| \leqslant \frac{1}{\sigma_{\min}(Z_J)} \leqslant O(1/\sigma_{\min}(Z))$. Therefore this in turn implies that $\|z_J\|_\infty \leqslant \|Z_J\|_{2\to\infty}\|\beta\| \leqslant O(\mu\sqrt{r/d}/\sigma_{\min}(Z)) \leqslant O(\mu\kappa\sqrt{r/d})$.

We will plug in $V = z_J v^T$ in the 2nd order optimality condition (5.3). Note that since $z_J \in$ col-span$(Z_J)$, it is supported on subset $J$, and therefore $\nabla^2 R(X)V = 0$. Therefore the term about regularization in (5.3) will vanish. For simplicity, let $y = X^\top z_J$, $w = Xv$ We obtain that taking $V = z_J v^\top$ in equation (5.3) will result in

$$\left\|P_\Omega(wz_J^\top + z_J w^\top)\right\|_F^2 \geqslant 2\langle P_\Omega(ZZ^\top - XX^\top), z_J z_J^\top\rangle$$

Note that we have that $\|w\|_\infty \leqslant \|X\|_{2\to\infty}\|v\| \leqslant \mu\sqrt{r/d}$. Recalling that $\|z_J\|_\infty \leqslant O(\mu\kappa\sqrt{r/d})$, by Theorem D.1, we have that

$$p\left\|wz_J^\top + z_J w^\top\right\|_F^2 \geqslant 2p\langle ZZ^\top - XX^\top, z_J z_J^\top\rangle - \delta p$$

where $\delta = O(\mu^2\kappa r^2(pd)^{-1/2})$. Then simple algebraic manipulation gives that

$$\langle w, z_J\rangle^2 + \|w\|^2\|z_J\|^2 + \|X^\top z_J\|^2 \geqslant \|Z^\top z_J\|^2 - \delta/2 \tag{5.14}$$

Note that $\langle w, z_J\rangle = \langle v, X^\top z_J\rangle = \langle y, v\rangle$. Recall that $\|z_J\| = 1$ and $z \in$ col-span$(Z_J)$, and therefore $\|Z^\top z_J\| = \|Z_J^\top z_J\| \geqslant \sigma_{\min}^2(Z_J)$. Moreover, recall that $\|y\| = \|X^\top z_J\| \leqslant \sigma_{\min}(X)$. Using these with equation (5.14) we obtain that

$$\begin{aligned}
\langle w, z_J\rangle^2 + \|w\|^2\|z_J\|^2 + \|X^\top z_J\|^2 &\leqslant \langle y, v\rangle^2 + \|w\|^2 + \|y\|^2 \\
&\leqslant 2\|y\|^2 + \sigma_{\min}^2(X) \qquad \text{(by Cauchy-Schwarz and } \|w\| = \sigma_{\min}(X).\text{)} \\
&\leqslant 3\sigma_{\min}^2(X) \qquad\qquad\qquad \text{(by } \|y\| \leqslant \sigma_{\min}(X).\text{)}
\end{aligned}$$

Therefore together with equation (5.14) and $\|Z^\top z_J\| \geqslant \sigma_{\min}^2(Z_J)$ we obtain that

$$\sigma_{\min}(X) \geqslant (1/2 - \Omega(\delta))\sigma_{\min}(Z_J) \tag{5.15}$$

Therefore combining equation (5.15) and the lower bound on $\sigma_{\min}(Z_J)$ we complete the proof. $\qquad\square$

Similar as before, we show it is possible to remove the regularizer term here, again the intuition is that the regularizer is always in the same direction as $X$.

**Lemma 5.7.** *Suppose $X$ satisfies equation* (5.4) *and* (5.13) *and* (5.10), *then for any $\gamma \geqslant 0$,*

$$\left\|ZZ^T X - XX^T X\right\|_F^2 \leqslant \left\|ZZ^T X - XX^T X - \gamma\nabla R(X)\right\|_F^2 \tag{5.16}$$

*Proof.* Let $L = \{i : \|X_i\| \geqslant \alpha\}$. For $i \notin L$, we have that $(\nabla R(X))_i = 0$. Therefore it suffices to prove that for every $i \in L$,

$$\left\|Z_i Z^\top X - X_i X^\top X\right\|^2 \leqslant \left\|Z_i Z^\top X - X_i X^\top X - (\gamma\nabla R(X))_i\right\|^2$$

It suffices to prove that

$$\langle(\nabla R(X))_i, X_i X^\top X - Z_i Z^\top X\rangle \geqslant 0 \tag{5.17}$$

By proposition 5.2, we have $\nabla R(X))_i = \Gamma_{ii}X_i$ for $\Gamma_{ii} \geqslant 0$. Then

$$\begin{aligned}
\langle(\nabla R(X))_i, X_i X^\top X\rangle &= \Gamma_{ii}\langle X_i, X_i X^\top X\rangle \\
&\geqslant \Gamma_{ii}\|X_i\|^2\sigma_{\min}(X^T X) \\
&\geqslant \frac{1}{4}\Gamma_{ii}\|X_i\|^2\sigma_{\min}(Z) \qquad \text{(by equation 5.13)}
\end{aligned}$$

On the other hand, we have

$$\langle(\nabla R(X))_i, Z_i Z^\top X\rangle = \Gamma_{ii}\langle X_i, Z_i Z^\top X\rangle$$

$$\leqslant \Gamma_{ii}\|X_i\|\|Z_i\|\sigma_{\max}(Z^T X) \leqslant \Gamma_{ii}\|X_i\|\|Z_i\|\sigma_{\max}(Z)\sigma_{\max}(X)$$
$$\leqslant \Gamma_{ii}\|X_i\|\|Z_i\|\sigma_{\max}(Z)^2 r^{1/6} \qquad\qquad \text{(by equation (5.10))}$$
$$\leqslant \frac{1}{10}\Gamma_{ii}\|X_i\|^2\sigma_{\max}(Z)^2 r^{-1/3} \qquad\qquad \text{(by } \|X_i\| \geqslant \alpha \geqslant 10\sqrt{r}\|Z_i\|)$$

Therefore combining two equations above we obtain equation (5.17) which completes the proof. $\qquad\square$

Finally we show the form in Equation (5.16) implies $ZZ^T$ is close to $XX^T$ (this is similar to Lemma 4.6).

**Lemma 5.8.** *Suppose $X$ and $Z$ satisfies that $\sigma_{\min}(X) \geqslant 1/4 \cdot \sigma_{\min}(Z)$ and that*

$$\left\|ZZ^T X - XX^T X\right\|_F^2 \leqslant \delta^2$$

*where $\delta \leqslant \sigma_{min}^3(Z)/C$ for a large enough constant $C$, then*

$$\|XX^\top - ZZ^\top\|_F^2 \leqslant O(\delta\kappa^2/\sigma_{min}(Z)).$$

*Proof.* The proof is similar to the one-dimensional case, we will separate $Z$ into the directions that are in column span of $X$ and its orthogonal subspace. We will then show the projection of $Z$ in the column span is close to $X$, and the projection on the orthogonal subspace must be small.

Let $Z = U + V$ where $U = \text{Proj}_{span(X)}Z$ is the projection of $Z$ to the column span of $X$, and $V$ is the projection to the orthogonal subspace. Then since $V^T X = 0$ we know

$$ZZ^T X = (U+V)(U+V)^T X = UU^T X + VU^T X.$$

Here columns of the first term $UU^T X$ are in the column span of $X$, and the columns second term $VU^T X$ are in the orthogonal subspace. Therefore,

$$\|ZZ^T X - XX^T X\|_F^2 = \|UU^T X - XX^T X\|_F^2 + \|VU^T X\|_F^2 \leqslant \delta^2.$$

In particular, both terms should be bounded by $\delta^2$. Therefore $\|UU^T - XX^T\|_F^2 \leqslant \delta^2/\sigma_{min}^2(X) \leqslant 16\delta^2/\sigma_{min}^2(Z)$.
Also, we know $\sigma_{min}(UU^T X) \geqslant \sigma_{min}(XX^T X) - \delta \geqslant \sigma_{min}(Z)^3/128$ if $\delta \leqslant \sigma_{min}(Z)^3/128$. Therefore $\sigma_{min}(U^T X)$ is at least $\sigma_{min}(Z)^3/\|Z\|128$. Now $\|V\|_F^2 \leqslant \delta^2/\sigma_{min}(U^T X)^2 \leqslant O(\delta^2\|Z\|^2/\sigma_{min}(Z)^6)$.
Finally, we can bound $\|UV^T\|_F$ by $\|U\|\|V\|_F \leqslant \|Z\|\|V\|_F$ (last inequality is because $U$ is a projection of $Z$), which at least $\Omega(\|V\|_F^2)$ when $\delta \leqslant \sigma_{min}(Z)^3/128$, therefore

$$\|ZZ^T - XX^T\|_F \leqslant \|UU^T - XX^T\|_F + 2\|UV^T\|_F + \|VV^T\|_F \leqslant O(\delta\|Z\|^2/\sigma_{min}(Z)^3).$$

$\qquad\square$

Last thing we need to prove the main theorem is a result from Sun and Luo[SL15], which shows whenever $XX^T$ is close to $ZZ^T$, the function is essentially strongly convex, and the only points that have 0 gradient are points where $XX^T = ZZ^T$, this is explained in Lemma C.1. Now we are ready to prove Theorem 5.3:

*Proof of Theorem 5.3.* Suppose $X$ satisfies 1st and 2nd order optimality condition. Then by Lemma 5.5 and Lemma 5.4, we have that $X$ satisfies equation (5.4), (5.9), (5.10) and (5.11). Then by Lemma 5.6, we obtain that $\sigma_{\min}(X) \geqslant 1/6 \cdot \sigma_{\min}(Z)$. Now by Lemma 5.7 and equation (5.11), we have that $\left\|ZZ^T X - XX^T X\right\|_F \leqslant \delta$ for $\delta \leqslant c\sigma_{\min}(Z)^3/\kappa^2$ for sufficiently small constant $c$. Then by Lemma 5.8 we obtain that $\|ZZ^\top - XX^\top\|_F \leqslant c\sigma_{\min}(Z)^2$ for sufficiently small constant $c$. By Lemma C.1, in this region the only points that satisfy the first order optimality condition must satisfy $XX^T = ZZ^T$. $\qquad\square$

**Handling Noise.** To handle noise, notice that we can only hope to get an approximate solution in presence of noise, and to get that our Lemmas only depend on concentration bounds which still apply in the noisy setting. See Section B for details.

# 6 Conclusions

Although the matrix completion objective is non-convex, we showed the objective function has very nice properties that ensures the local minima are also global. This property gives guarantees for many basic optimization algorithms. An important open problem is the robustness of this property under different model assumptions: Can we extend the result to handle asymmetric matrix completion? Is it possible to add weights to different entries (similar to the settings studied in [LLR16])? Can we replace the objective function with a different distance measure rather than Frobenius norm (which is related to works on 1-bit matrix sensing [DPvdBW14])? We hope this framework of analyzing the geometry of objective function can be applied to other problems.

## Footnotes

[2]The entries $(i, j)$ and $(j, i)$ are the same. With probability $p$ we observe both entries and otherwise we observe neither.

[3] This is the main reason for us to choose 4-th power instead of 2-nd power.

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

# A  Omitted Proofs in Section 4

We first prove the equivalent form of the first and second order optimality conditions:

**Lemma A.1** (Proposition 4.1 restated). *The first order optimality condition of objective* (4.1) *is,*

$$2P_\Omega(M - xx^\top)x = \lambda \nabla R(x) \,,$$

*and the second order optimality condition requires:*

$$\forall v \in \mathbb{R}^d, \; \|P_\Omega(vx^\top + xv^\top)\|_F^2 + \lambda v^\top \nabla^2 R(x)v \geqslant 2v^\top P_\Omega(M - xx^\top)v \,.$$

*Moreover, The $\tau$-relaxed second order optimality condition requires*

$$\forall v \in \mathbb{R}^d, \; \|P_\Omega(vx^\top + xv^\top)\|_F^2 + \lambda v^\top \nabla^2 R(x)v \geqslant 2v^\top P_\Omega(M - xx^\top)v - \tau \|v\|^2 \,.$$

*Proof.* We take the Taylor's expansion around point $x$. Let $\delta$ be an infinitesimal vector, we have

$$
\begin{aligned}
f(x + \delta) &= \frac{1}{2}\|P_\Omega(M - (x + \delta)(x + \delta)^\top)\|_F^2 + \lambda R(x + \delta) + o(\|\delta\|^2) \\
&= \frac{1}{2}\|P_\Omega(M - xx^\top - (x\delta^\top + \delta x^\top) - \delta\delta^\top)\|_F^2 + \lambda\left(R(x) + \langle \nabla R(x), \delta\rangle + \frac{1}{2}\delta^T \nabla^2 R(x)\delta\right) + o(\|\delta\|^2) \\
&= \frac{1}{2}\|M - xx^\top\|_\Omega^2 + \lambda R(x) \\
&\quad - \langle P_\Omega(M - xx^\top), x\delta^\top + \delta x^\top\rangle + \langle \nabla R(x), \delta\rangle + o(\|\delta\|^2) \\
&\quad - \langle P_\Omega(M - xx^\top), \delta\delta^\top\rangle + \frac{1}{2}\|P_\Omega(x\delta^\top + \delta x^\top)\|_F^2 + \frac{1}{2}\lambda\delta^\top \nabla^2 R(x)\delta + o(\|\delta\|^2).
\end{aligned}
$$

By symmetry $\langle P_\Omega(M - xx^\top), x\delta^\top\rangle = \langle P_\Omega(M - xx^\top), \delta x^\top\rangle = \langle P_\Omega(M - xx^\top)x, \delta\rangle$, so the first order optimality condition is $\forall \delta, \langle -2P_\Omega(M - xx^\top)x + \lambda \nabla R(x), \delta\rangle = 0$, which is equivalent to that $2P_\Omega(M - xx^\top)x = \lambda \nabla R(x)$.

The second order optimality condition says $-\langle P_\Omega(M - xx^\top), \delta\delta^\top\rangle + \frac{1}{2}\|x\delta^\top + \delta x^\top\|_F^2 + \frac{1}{2}\lambda\delta^\top \nabla^2 R(x)\delta \geqslant 0$ for every $\delta$, which is exactly equivalent to Equation (4.3). $\qquad\square$

Next we show the full proof for the second order optimality condition:

**Lemma A.2** (Lemma 4.8 restated). *In the setting of Theorem 4.2, with high probability over the choice of $\Omega$, suppose $x \in \mathcal{B}'$ satisfies second-order optimality condition* (4.3) *or $\tau$-relaxed condition for $\tau \leqslant 0.1p$, we have $\|x\|^2 \geqslant 1/8$.*

*Proof.* If $\|x\| \geqslant 1$, then we are done. Therefore in the rest of the proof we assume $\|x\| \leqslant 1$. The proof is very similar to Lemma 4.4. We plug in $v = z_J$ instead into equation (4.3), where $J = \{i : |x_i| \leqslant \alpha\}$. Note that $\nabla R(z_J)$ vanishes. We plug in $v = z_J$ in the equation (4.3) and obtain that $x$ satisfies that

$$\left\|P_\Omega(z_J x^\top + x z_J^\top)\right\|_F^2 \geqslant 2z_J^\top P_\Omega(M - xx^\top)z_J \,. \tag{A.1}$$

Note that we assume $\|x\|_\infty \leqslant 2\alpha$, and in the beginning of the proof we assume wlog $\|x\| \leqslant 1$. Moreover, we have $\|z_J\| \leqslant \frac{\mu}{\sqrt{d}}$ an, $\|z_J\| \leqslant 1$. Similarly to the derivation in the proof of Lemma 4.4, we apply Theorem D.1 (twice) and obtain that with high probability over the choice of $\Omega$, for every $x$, for $\varepsilon = \tilde{O}(\mu^2(pd)^{-1/2})$,

$$\text{LHS of (A.1)} = p\left\|z_J x^\top + x z_J^\top\right\|_F^2 \pm O(p\varepsilon) = 2p\|x\|^2\|z_J\|^2 + 2p\langle x, z_J\rangle^2 \pm O(p\varepsilon) \,.$$

$$
\begin{aligned}
\text{RHS of (A.1)} &= 2\left(\langle P_\Omega(zz^\top), P_\Omega(z_J z_J^\top)\rangle - \langle P_\Omega(xx^\top), P_\Omega(z_J z_J^\top)\rangle\right) &&\text{(Since } M = zz^\top\text{)} \\
&= 2\|z_J\|^4 - 2\langle x, z_J\rangle^2 \pm O(p\varepsilon) \,. &&\text{(by Theorem D.1)}
\end{aligned}
$$

(Again notice that using $\tau$-relaxed second order optimality condition does not effect the RHS by too much, so it does not change later steps.) Therefore plugging the estimates above back into equation (A.1), we have that

$$p\|x\|^2\|z_J\|^2 + 2p\langle x, z_J\rangle^2 \geqslant p\|z_J\|^4 \pm O(p\varepsilon),$$

Using Cauchy-Schwarz, we have $\|x\|^2\|z_J\|^2 \geqslant \langle x, z_J\rangle^2$, and therefore we obtain that $\|z_J\|^2\|x\|^2 \geqslant \frac{1}{3}\|z_J\|^4 - O(\varepsilon)$.

Finally, we claim that $\|z_J\|^2 \geqslant 1/2$, which completes the proof since $\|x\|^2 \geqslant \frac{1}{3}\|z_J\|^2 - O(\varepsilon) \geqslant 1/8$.

*Claim* A.3. Suppose $\alpha \geqslant \frac{4\mu}{\sqrt{d}}$ and $x$ satisfies $\|x\|_\infty \leqslant 4\alpha$ and $\|x\| \leqslant 2$. Let $J = \{i : |x_i| \leqslant \alpha\}$. Then we have that $\|z_J\| \geqslant 1/2$.

The claim can be simply proved as follows: Since $\|x\|^2 \leqslant 2$ we have that $|J^c| \leqslant 2/\alpha^2$ and therefore $\|z_{J^c}\|^2 \leqslant 2\mu^2/(d\alpha^2)$. This further implies that $\|z_J\|^2 = \|z\|^2 - \|z_L\|^2 \geqslant (1 - 2\mu^2/(d\alpha^2)) \geqslant \frac{1}{2}$ because $\alpha \geqslant \frac{2\mu}{\sqrt{d}}$. $\qquad\square$

**Lemma A.4** (Lemma 4.10 restated). *Suppose $x \in \mathcal{B}'$ satisfies that $\|x\|^2 \geqslant 1/8$, under the same assumption as Lemma 4.9. we have,*

$$\big\|\langle x, z\rangle z - \|x\|^2 x\big\| \leqslant O(\varepsilon)$$

*Proof.* Let $L = \{i : \|x_i\| \geqslant \alpha\}$. For $i \notin L$, we have that $(\nabla R(x))_i = 0$. Therefore it suffices to prove that for every $i \in L$,

$$(z_i z^\top x - x_i\|x\|)^2 \leqslant (z_i z^\top x - x_i\|x\| - (\gamma\nabla R(x))_i)^2$$

It suffices to prov that

$$(\nabla R(x))_i(x_i\|x\|^2 - z_i\langle z, x\rangle) \geqslant 0 \qquad\qquad\qquad (\text{A.2})$$

Since we have $\nabla R(x)_i = \gamma_i x_i$ for some $\gamma_i \geqslant 0$, we have

$$\begin{aligned}(\nabla R(x))_i \cdot x_i\|x\|^2 &= \langle\gamma_i x_i, x_i\|x\|^2\rangle \\ &\geqslant \gamma_i x_i^2\|x\|^2 \\ &\geqslant \frac{1}{\sqrt{8}}\gamma_i x_i^2\|x\| \qquad\qquad (\text{since } \|x\|^2 \geqslant 1/8)\end{aligned}$$

On the other hand, we have

$$\begin{aligned}(\nabla R(x))_i \cdot z_i\langle z, x\rangle &= \gamma_i x_i z_i\langle z, x\rangle \\ &\leqslant \frac{1}{4}\gamma_i x_i^2\|x\|\|z\| \qquad\qquad (\text{by } |x_i| \geqslant \alpha \geqslant 4|z_i|)\end{aligned}$$

Therefore combining two equations above we obtain equation (A.2) which completes the proof. $\qquad\square$

# B   Handling Noise

Suppose instead of observing the matrix $ZZ^T$, we actually observe a noisy version $M = ZZ^T + N$, where $N$ is a Gaussian matrix with independent $N(0, \sigma^2)$ entries. In this case we should not hope to exactly recover $ZZ^T$ (as two close $Z$'s may generate the same observation). In this Section we show even with fairly large noise our arguments can still hold.

**Theorem B.1.** *Let $\hat{\mu} = \max\{\mu, \sqrt{\frac{4\sigma d\sqrt{\log d}}{r}}\}$. Suppose $p \geqslant C\hat{\mu}^6\kappa^{12}r^4 d^{-1}\varepsilon^{-2}\log^{1.5} d$ where $C$ is a large enough constant. Let $\alpha = 2\hat{\mu}\kappa r/\sqrt{d}, \lambda \geqslant \hat{\mu}^2 rp/\alpha^2$. Then with high probability over the randomness of $\Omega$, any local minimum $X$ of $f(\cdot)$ satisfies*

$$\|XX^T - ZZ^T\|_F \leqslant \varepsilon.$$

*In fact, a noise level $\sigma\sqrt{\log d} \leqslant \mu^2 r/d$ (when the noise is almost as large as the maximum possible entry) does not change the conclusions of Lemmas in this Section.*

*Proof.* There are only three places in the proof where the noise will make a difference. These are: 1. The infinity norm bound of $M$, used in Lemma 5.4. 2. The LHS of first order optimality condition (Equation (5.2)). 3. The RHS of second order optimality condition (Equation (5.3)).

What we require in these three steps are: 1. $|M|_\infty$ should be smaller than $\mu^2 r/d$. 2. $\langle P_\Omega(N), W \rangle$ should be smaller than $|\langle P_\Omega(N), P_\Omega(W) \rangle| \leqslant O(\sigma |Z|_\infty dr \log d + \sqrt{pd^2 r\sigma^2 |W|_\infty \|W\|_F \log d})$. 3. $\|P_\Omega(N)\| \leqslant \varepsilon p \|ZZ^T\|_F$. When we define the $\hat{\mu} = \max\{\mu, \sqrt{\frac{4\sigma d \sqrt{\log d}}{r}}\}$, all of these are satisfied (by Lemma D.5 and D.6).

Now we can follow the proof and see $\delta \leqslant c\varepsilon\sigma_{\min}(Z)/\kappa^2$ for small enough constant $c$, and By Lemma 5.8 we know $\|XX^T - ZZ^T\|_F \leqslant \varepsilon$. $\qquad \square$

# C   Finding the Exact Factorization

In Section 5, we showed that any point that satisfies the first and second order necessary condition must satisfy $\|XX^T - ZZ^T\|_F \leqslant c$ for a small enough constant $c$. In this section we will show that in fact $XX^T$ must be exactly equal to $ZZ^T$. The proof technique here is mostly based on the work of Sun and Luo[SL15]. However we have to modify their proof because we use slightly different regularizers, and we work in the symmetric case. The main Lemma in [SL15] can be rephrased as follows in our setting:

**Lemma C.1** (Analog to Lemma 3.1 in [SL15]). *Suppose $p \geqslant C\mu^4 r^6 \kappa^4 d^{-1} \log d$ for large enough absolute constant $C$, and $\varepsilon = \sigma_{min}(Z)^2/100$. with high probability over the randomness of $\Omega$, we have that for any point $X$ in the set*

$$\mathcal{B}_\varepsilon = \left\{ X \in \mathbb{R}^{d\times r} : \|XX^T - ZZ^T\|_F \leqslant \varepsilon, \|X\|_{2\to\infty} \leqslant \frac{16\mu\kappa r}{\sqrt{d}} \right\}, \qquad (C.1)$$

*there exists a matrix $U$ such that $UU^T = ZZ^T$ and*

$$\langle \nabla f(X), X - U \rangle \geqslant \frac{p}{4}\|M - XX^T\|_F^2.$$

*As a consequence, any point $X$ in the set $\mathcal{B}$ that satisfies first order optimality condition must be a global optimum (or, equivalently, satisfy $XX^T = ZZ^T$).*

Recall $f(X) = \frac{1}{2}\|P_\Omega(M - XX^T)\|_F^2 + \lambda R(X)$. The proof of Lemma C.1 consists of three steps:

1. The regularizer has nonnegative correlation with $(X - U)$: for any $U$ such that $UU^T = ZZ^T$, we have $\langle \nabla R(X), X - U \rangle \geqslant 0$. (Claim C.3).

2. There exists a matrix $U$ such that $UU^T = ZZ^T$, and $U$ is close to $X$. (Claim C.4)

3. Argue that $\langle \nabla f(x), X - U \rangle \geqslant \frac{p}{4}\|P_\Omega(M - XX^T)\|_F^2$ when $U$ is close to $X$. (See proof of Lemma C.1).

Before going into details, the first useful observation is that all matrices $U$ with $UU^T = ZZ^T$ have the same row norm.

*Claim* C.2. Suppose $U, Z \in \mathbb{R}^{d\times r}$ satisfy $UU^\top = ZZ^\top$. Then, for any $i \in [d]$ we have $\|U_i\| = \|Z_i\|$. Consequently, $\|U\|_F = \|Z\|_F$.

*Proof.* Suppose $UU^\top = ZZ^\top$, then we have $U = ZR$ where $R$ is an orthonormal matrix. In particular, the $i$-th row of $U$ is equal to

$$U_i = Z_i R.$$

Since $\ell_2$ norm (and Frobenius norm) is preserved after multiplying with an orthonormal matrix, we know $\|U_i\| = \|Z_i\|$. The Frobenius norm bound follows immediately. $\qquad \square$

Note that this simple observation is only true in the symmetric case. This Claims serves as the same role of the bounds on row norms of $U, V$ in the asymmetric case (Propositions 4.1 and 4.2 of [SL15]).

Next we are ready to argue that the regularizer is always positively correlated with $X - U$.

*Claim* C.3. For any $U$ such that $UU^T = ZZ^T$, we have,

$$\langle \nabla R(X), X - U \rangle \geqslant 0.$$

*Proof.* Since the regularizer is applied independently to individual rows, we can rewrite $\langle \nabla R(X), X - U \rangle = \sum_{i=1}^{n} \langle \nabla R(X_i), X_i - U_i \rangle$, and focus on $i$-th row.

For each row $X_i$, $\nabla R(X_i)$ is 0 when $\|X_i\| \leqslant 2\mu\sqrt{r}/\sqrt{d}$. In that case $\langle \nabla R(X_i), X_i - U_i \rangle = 0$.

When $\|X_i\|$ is larger than $2\mu/\sqrt{d}$, we know $\nabla R(X_i)$ is always in the same direction as $X_i$. In this case $\lambda \nabla R(X_i) = \gamma X_i$ for some $\gamma > 0$ and $\|X_i\| \geqslant 2\mu\sqrt{r}/\sqrt{d} \geqslant 2\|Z_i\| = 2\|U_i\|$ (where last equality is by Claim C.2). Therefore by triangle inequality

$$\langle X_i, X_i - U_i \rangle \geqslant \|X_i\|^2 - \|X_i\|\|U_i\| \geqslant \|X_i\|^2/2 > 0.$$

This then implies $\langle \lambda \nabla R(X_i), X_i - U_i \rangle = \gamma \langle X_i, X_i - U_i \rangle > 0$.

$\square$

Next we will prove the gradient of $\frac{1}{2}\|P_\Omega(M - XX^T)\|_F^2$ has a large correlation with $X - U$. This is analogous to Proposition 4.2 in [SL15].

**Claim C.4.** Suppose $\|XX^T - M\|_F = \varepsilon \leqslant \sigma_{min}(Z)^2/100$, there exists a matrix $U$ such that $UU^T = M$ and $\|X - U\|_F \leqslant 5\varepsilon\sqrt{r}/\sigma_{min}(Z)^2$.

*Proof.* Without loss of generality we assume $M$ is a diagonal matrix with first $r$ diagonal terms being $\sigma_1(Z)^2, \sigma_2(Z)^2, ..., \sigma_r(Z)^2$ (this can be done by a change of basis). That is, we assume $M = \mathbf{diag}(\sigma_1(Z)^2, \ldots, \sigma_r(Z)^2, 0, \ldots, 0)$. We use $M'$ to denote the first $r \times r$ principle submatrix of $M$.

We write $X = \begin{bmatrix} V \\ W \end{bmatrix}$ where $V$ contains the first $r$ rows of $X$, and $W \in \mathbb{R}^{(d-r) \times r}$ contains the remaining rows in $X$.

We write similarly $U = \begin{bmatrix} P \\ Q \end{bmatrix}$ where $P$ and $Q$ denote the first $r$ rows and the rest of rows respectively.

In order to construct $U$, we first notice that $Q$ must be constructed as a zero matrix since $M$ has non-zero diagonal only on the top-left corner. A natural guess of $P$ then becomes a "normalized" version of $V$.

Concretely, we construct $P := VS = V(V^T(M')^{-1}V)^{-1/2}$ (where $S := (V^T(M')^{-1}V)^{-1/2}$). Thus, the difference between $U$ and $X$ is equal to $\|U - X\|_F \leqslant \|P - V\|_F + \|W\|_F$.

Since $\|XX^T - M\|_F \leqslant \varepsilon$, we know $\|M' - VV^T\|_F^2 + 2\|VW^T\|_F^2 \leqslant \varepsilon^2$. In particular both terms are smaller than $\varepsilon^2$.

First, we bound $\|W\|_F$. Note that since $\|M' - VV^T\|_F \leqslant \varepsilon \leqslant \sigma_{min}(Z)^2/100$, we know $\sigma_{min}(V)^2 \geqslant 0.99\sigma_{min}(Z)^2$. Therefore $\sigma_{min}(V) \geqslant 0.9\sigma_{min}(Z)$. Now we know $\|W\|_F \leqslant \|VW^T\|_F/\sigma_{min}(V) \leqslant 2\varepsilon/\sigma_{min}(Z)$.

Next we bound $\|P - V\|_F^2$. Since $\|M' - VV^T\|_F \leqslant \varepsilon \leqslant \sigma_{min}(Z)^2/100$, we know $(1 - 2\varepsilon/\sigma_{min}(Z)^2)VV^T \preceq M' \preceq (1 + 2\varepsilon^2/\sigma_{min}(Z)^2)VV^T$. This implies $\|V\|_F \leqslant 1.1\|Z\|_F$, and $(1 - 2\varepsilon/\sigma_{min}(Z)^2)I \preceq V^TM^{-1}V \preceq (1 + 2\varepsilon/\sigma_{min}(Z)^2)I$. Therefore the matrix $S$ is also very close to identity, in particular, $\|S - I\| \leqslant 2\varepsilon/\sigma_{min}(Z)^2$. Now we know $\|P - V\|_F = \|V\|_F\|S - I\| \leqslant 3\varepsilon\|Z\|_F/\sigma_{min}(Z)^2$. Using the fact that $\|Z\|_F = 1$ we know $\|U - X\|_F \leqslant \|P - V\|_F + \|W\|_F \leqslant 5\varepsilon\sqrt{r}/\sigma_{min}(Z)^2$.

$\square$

We can now combine this Claim with a sampling lemma to show $\|P_\Omega((X - U)(X - U)^T)\|_F^2$ is small:

**Lemma C.5.** *Under the same setting of Lemma C.1, with probability at least $1 - 1/(2n)^4$ over the choice of $\Omega$, if $U$ satisfies conclusion of Claim C.4, then,*

$$\|P_\Omega((X - U)(X - U)^T)\|_F^2 \leqslant \frac{p}{25}\|M - XX^T\|_F^2.$$

Intuitively, this Lemma is true because $\|(X - U)(X - U)^T\|_F \leqslant 25\|M - XX^T\|_F^2 r/\sigma_{min}(Z)^4$, which is much smaller than $\|M - XX^T\|_F$ when $\|M - XX^T\|_F$ is small. By concentration inequalities we expect $\|P_\Omega((X-U)(X-U)^T)\|_F^2$ to be roughly equal to $p\|(X - U)(X - U)^T\|_F$, therefore it must be much smaller than $p\|M - XX^T\|_F^2$. The proof of this Lemma is exactly the same as Proposition 4.3 in [SL15] (in fact, it is directly implied by Proposition 4.3), so we omit the proof here. We also need a different concentration bound for the projection of the norm of the matrix $a = U(X - U)^T + (X - U)U^T$. Unlike the previous lemma, here we want $\|P_\Omega(a)\|_F$ to be large.

**Lemma C.6.** *Under the same setting of Lemma C.1, let $a = U(X - U)^T + (X - U)U^T$ where $U$ is constructed as in Claim C.4. Then, with high probability, we have that for any $X \in \mathcal{B}_\varepsilon$,*

$$\|P_\Omega(a)\|_F^2 \geqslant \frac{5p}{6}\|a\|_F^2.$$

Intuitively this should be true because $a$ is in the tangent space $\{Z : Z = UW^T + (W')U^T\}$ which has rank $O(nr)$. The proof of this follows from Theorem 3.4 [Rec11], and is written in detail in Equations (37) - (41) in [SL15].

Finally we are ready to prove the main lemma. The proof is the same as the outline given in Section 4.1 of [SL15]. We give it here for completeness.

*Proof of Lemma C.1.* Note that $f(X)$ is equal to $h(X) + \lambda R(X)$ where where $h(X) = \frac{1}{2}\|P_\Omega(M - XX^T)\|_F^2$, and $R(X)$ is the regularizer. By Claim C.3 we know $\langle \nabla R(X), X - U \rangle \geqslant 0$, so we only need to prove there exists a $U$ such that $UU^T = Z$ and $\langle \nabla g(X), X - U \rangle \geqslant \frac{p}{4}\|M - XX^T\|_F^2$.

Define $a = U(X - U)^T + (X - U)U^T$, $b = (U - X)(U - X)^T$, then $XX^T - M = a + b$ and $(X - U)X^T + X(X - U)^T = a + 2b$.

Now

$$\begin{aligned}
\langle \nabla h(X), X - U \rangle &= 2\langle P_\Omega(XX^T - M)X, X - U \rangle \\
&= \langle P_\Omega(XX^T - M), (X - U)X^T + X(X - U)^T \rangle \\
&= \langle P_\Omega(a + b), P_\Omega(a + 2b) \rangle \\
&= \|P_\Omega(a)\|_F^2 + 2\|P_\Omega(b)\|_F^2 + 3\langle P_\Omega(a), P_\Omega(b) \rangle \\
&\geqslant \|P_\Omega(a)\|_F^2 + 2\|P_\Omega(b)\|_F^2 - 3\|P_\Omega(a)\|\|P_\Omega(b)\|.
\end{aligned}$$

Let $\varepsilon = \|M - XX^T\|_F$. Note that from Claim C.4 and Lemma C.5, we know

$$\|b\|_F \leqslant \varepsilon/10, \quad \|P_\Omega(b)\|_F \leqslant \sqrt{p}d/5.$$

Therefore as long as we can show $\|P_\Omega(a)\|_F$ is large we are done. This is true because $\|a\|_F \geqslant \|M - XX^T\|_F - \|b\|_F \geqslant 9\varepsilon/10$. Hence by Lemma C.6 we know

$$\|P_\Omega(A)\|_F^2 \geqslant \frac{5p}{6}\|a\|_F^2 \geqslant \frac{27}{40}p\varepsilon^2.$$

Combining the bounds for $\|P_\Omega(a)\|_F$, $\|P_\Omega(b)\|_F$, we know $\langle \nabla g(X), X - U \rangle \geqslant \frac{p}{4}\|M - XX^T\|_F^2$. Together with the fact that $\langle \nabla R(X), X - U \rangle \geqslant 0$, we know

$$\langle \nabla f(X), X - U \rangle \geqslant \frac{p}{4}\|M - XX^T\|_F^2.$$

$\square$

# D  Concentration inequality

In this section we prove the concentration inequalities used in the main part. We first show that the inner-product of two low rank matrices is preserved after restricting to the observed entries. This is mostly used in arguments about the second order necessary conditions.

**Theorem D.1.** *With high probability over the choice of $\Omega$, for any two rank-$r$ matrices $W, Z \in \mathbb{R}^{d \times d}$, we have*

$$|\langle P_\Omega(W), P_\Omega(Z) \rangle - p\langle W, Z \rangle| \leqslant O(|W|_\infty |Z|_\infty dr \log d + \sqrt{pdr|W|_\infty |Z|_\infty \|W\|_F \|Z\|_F \log d})$$

*Proof.* Since both LHS and RHS are bilinaer in both $W$ and $Z$, without loss of generality we assume the Frobenius norms of $W$ and $Z$ are all equal to 1. Note that in this case we should expect $|W|_\infty \geqslant 1/d$.

Let $\delta_{i,j}$ be the indicator variable for $\Omega$, we know

$$\langle P_\Omega(W, Z) = \sum_{i,j} \delta_{i,j} W_{i,j} Z_{i,j},$$

and in expectation it is equal to $p\langle W, Z \rangle$. Let $Q = \sum_{i,j}(\delta_{i,j} - p)W_{i,j}Z_{i,j}$. We can then view $Q$ as a sum of independent entries (note that $\delta_{i,j} = \delta_{j,i}$, but we can simply merge the two terms and the variance is at most a factor 2 larger). The expectation $\mathbb{E}[Q] = 0$. Each entry in the sum is bounded by $|W|_\infty |Z|_\infty$, and the variance is bounded by

$$\mathbb{V}[Q] \leqslant p \sum_{i,j} (W_{i,j} Z_{i,j})^2$$

$$\leqslant p \max_{i,j} |W_{i,j}|^2 \sum_{i,j} Z_{i,j}^2$$

$$\leqslant p|W|_\infty^2.$$

Similarly, we also know $\mathbb{V}[Q] \leqslant p|Z|_\infty^2$ and hence $\mathbb{V}[Q] \leqslant p|W|_\infty |Z|_\infty$.

Now we can apply Bernstein's inequality, with probability at most $\eta$,

$$|Q - \mathbb{E}[Q]| \geqslant |W|_\infty |Z|_\infty \log 1/\eta + \sqrt{p|W|_\infty |Z|_\infty \log(1/\eta)}.$$

By Proposition E.2, there is a set $\Gamma$ of size $d^{O(dr)}$ such that for any rank $r$ matrix $X$, there is a matrix $\hat{X} \in \Gamma$ such that $\|X - \hat{X}\|_F \leqslant 1/d^3$. When $W$ and $Z$ come from this set, we can set $\eta = d^{-Cdr}$ for a large enough constant $C$. By union bound, with high probability

$$|Q - \mathbb{E}[Q]| \leqslant O(|W|_\infty |Z|_\infty dr \log d + \sqrt{pdr|W|_\infty |Z|_\infty \log d}).$$

When $W$ and $Z$ are not from this set $\Gamma$, let $\hat{W}$ and $\hat{Z}$ be the closest matrix in $\Gamma$, then we know $|\langle P_\Omega(W), P_\Omega(Z) \rangle - p\langle W, Z \rangle - (\langle P_\Omega(\hat{W}), P_\Omega(\hat{Z}) \rangle - p\langle \hat{W}, \hat{Z} \rangle)| \leqslant O(1/d^3) \ll |W|_\infty |Z|_\infty dr \log d$. Therefore we still have

$$|\langle P_\Omega(W), P_\Omega(Z) \rangle - p\langle W, Z \rangle| \leqslant O(|W|_\infty |Z|_\infty dr \log d + \sqrt{pdr|W|_\infty |Z|_\infty \|W\|_F \|Z\|_F \log d}).$$

$\square$

Next Theorem shows $P_\Omega(XX^T)X$ is roughly equal to $pXX^TX$, this is one of the major terms in the gradient.

**Theorem D.2.** *When $p \geqslant \frac{C\nu^6 r \log^{1.5} d}{d\varepsilon^2}$ for a large enough constant $C$, With high probability over the randomness of $\Omega$, for any matrix $X \in \mathbb{R}^{d \times r}$ such that $\|X_i\| \leqslant \nu\sqrt{\frac{1}{d}}\|X\|_F$, we have*

$$\|P_\Omega(XX^\top)X - pXX^TX\|_F \leqslant p\varepsilon\|X\|_F^3 \tag{D.1}$$

*Proof.* Without loss of generality we assume $\|X\|_F = 1$. Let $\delta_{i,j}$ be the indicator variable for $\Omega$, we first prove the result when $\delta_{i,j}$ are independent, then we will use standard techniques to show the same argument works for $\delta_{i,j} = \delta_{j,i}$.

Note that

$$[P_\Omega(XX^\top)X]_i = \sum_j \delta_{i,j}\langle X_i, X_j \rangle X_j,$$

whose expectation is equal to

$$[pXX^TX]_i = p\sum_j \langle X_i, X_j \rangle X_j.$$

We know $\|X_i\| \leqslant \nu\sqrt{\frac{1}{d}}$, therefore each term is bounded by $\nu^3(1/d)^{3/2}$. Let $Z_i$ be a random variable that is equal to $\|P_\Omega(XX^\top)X]_i - [pXX^TX]_i\|^2$, then it is easy to see $\mathbb{E}[Z_i] \leqslant pd\nu^6(r/d)^3 = p\nu^6/d^2$. and the variance

$\mathbb{V}[Z_i] = \mathbb{E}[Z_i^2] - \mathbb{E}[Z_i]^2 \leqslant pd\nu^{12}(1/d)^6 + 2\,\mathbb{E}[Z_i]^2 \leqslant 3\,\mathbb{E}[Z_i]^2$ (as long as $p > 1/d$). Our goal now is to prove $\sum_{i=1}^{d} Z_i \leqslant p^2\varepsilon^2$ for all $X$.

Let $\overline{Z}_i$ be a truncated version of $Z_i$. That is, $\overline{Z}_i = Z_i$ when $Z_i \leqslant [2pd\nu^3(1/d)^{3/2}]^2$, and $\overline{Z}_i = [2pd\nu^3(1/d)^{3/2}]^2$ otherwise. It's not hard to see $\overline{Z}_i$ has smaller mean and variance compared to $Z_i$. Also, by vector's Bernstein's inequality, we know

$$\mathbb{P}[\sqrt{\overline{Z}_i} \leqslant t] \leqslant d\exp\left(-\frac{t^2}{\frac{p\nu^6}{d^2} + t\frac{\nu^3}{d^{3/2}}}\right).$$

Notice that this is only relevant when $t \leqslant O(p\nu^3 d^{-1/2})$ (because otherwise the probability is 0) and in that regime the variance term always dominates. Therefore $\overline{Z}_i$ is the square of a subgaussian random variable.

By the Bernstein's inequality, we know the moments of $\sqrt{\overline{Z}_i} - \mathbb{E}[\sqrt{\overline{Z}_i}]$ are dominated by a Gaussian distribution with variance $O(\mathbb{E}[\overline{Z}_i\sqrt{\log d})$.

Now we can use the concentration bound for quadratics of the subgaussian random variables[HKZ12]: we know that with probability $\exp(-t)$,

$$\sum_{i=1}^{d} \overline{Z}_i \leqslant O(\mathbb{E}[Z_i^2]\sqrt{\log d}(d + 2\sqrt{dt} + 2t)).$$

this means with probability $\exp(-Cdr\log d)$ with some large constant $C$, we know $\sum_{i=1}^{d} \overline{Z}_i \leqslant O(p\nu^6 r\log^{1.5} d/d)$. The probability is low enough for us to union bound over all $X$ in a standard $\varepsilon$-net such that every other $X$ is within distance $(\varepsilon/d)^6$. Therefore we know with high probability for all $X$ in the $\varepsilon$-net we have $\sum_{i=1}^{d} \overline{Z}_i \leqslant O(p\nu^6 r\log^{1.5} d/d)$, which is smaller than $p^2\varepsilon^2$ when $p \geqslant \frac{C\nu^6 r\log^{1.5} d}{d\varepsilon^2}$ for a large enough constant $C$.

For any $\hat{X}$ that is not in the $\varepsilon$-net, let $X$ be the closest point of $X$ in the net, then $\|X - \hat{X}\|_F \leqslant 1/d^6$, therefore the bound of $\hat{X}$ clearly follows from the bound of $X$.

Now to convert sum of $\overline{Z}_i$ to sum of $Z_i$, notice that with high probability there are at most $2pd$ entries in $\Omega$ for every row. When that happens $Z_i$ is always bounded by $[2pd\nu^3(1/d)^{3/2}]^2$ so $Z_i = \overline{Z}_i$. Let event 1 be $\sum_{i=1}^{d} \overline{Z}_i \leqslant p^2\varepsilon^2$ for all $X$, and let event 2 be that there are at most $2pd$ entries per row, we know with high probability both event happens, and in that case $\sum_{i=1}^{d} Z_i \leqslant p^2\varepsilon^2$ for all $X$.

**Handling $\delta_{i,j} = \delta_{j,i}$.** First notice that the diagonal entries $\delta_{i,i}$'s cannot change the Frobenius norm by more than $O(\nu^3(1/d)^{3/2} \cdot \sqrt{d}) \leqslant p\varepsilon$ so we can ignore the diagonal terms. Now for independent terms $\delta_{i,j}$, let $\gamma_{j,i} = \delta_{i,j}$, then by union bound both $\delta_{i,j}$ and $\gamma_{i,j}$ satisfy the equation, and by triangle's inequality $(\delta_{i,j} + \gamma_{i,j})/2$ also satisfies the inequality. Let $\tau_{i,j}$ be the true indicator of $\Omega$ (hence $\tau_{i,j} = \tau_{j,i}$), and $\tau'_{i,j}$ be an independent copy, we know $(\tau_{i,j} + \tau'_{i,j})/2$ has the same distribution as $(\delta_{i,j} + \gamma_{i,j})/2$ (for off-diagonal entries), therefore with high probability the equation is true for $(\tau_{i,j} + \tau'_{i,j})/2$. The Theorem then follows from the standard Claim below for decoupling (note that $\sup_{\|X\|_F=1} \|P_\Omega(XX^T)X - pXX^TX\|_F$ is a norm for the indicator variables of $\Omega$):

*Claim* D.3. Let $X, Y$ be two iid random variables, then

$$\mathbb{P}[\|X\| \geqslant t] \leqslant 3\,\mathbb{P}[\|X + Y\| \geqslant \frac{2t}{3}].$$

*Proof.* Let $X, Y, Z$ be iid random variables then,

$$\begin{aligned}
\mathbb{P}[X \geqslant t] &= \mathbb{P}[\|(X + Y) + (X + Z) - (Y + Z)\| \geqslant 2t]\\
&\leqslant \mathbb{P}[\|X + Y\| \geqslant 2t/3] + \mathbb{P}[\|X + Z\| \geqslant 2t/3] + \mathbb{P}[\|Y + Z\| \geqslant 2t/3]\\
&\leqslant 3\,\mathbb{P}[\|X + Y\| \geqslant \frac{2t}{3}].
\end{aligned}$$

$\square$

$\square$

Finally we argue that random sampling of a matrix gives a nice spectral approximation. This is a standard Lemma that is used in arguing about the $P_\Omega(M)X$ term in the gradient $(P_\Omega(M - XX^T)X)$.

**Lemma D.4.** *Suppose $W \in \mathbb{R}^{d \times d}$ satisfies that $|W|_\infty \leqslant \frac{\nu}{d}\|W\|_F$, then with high probability $(1 - d^{-10})$ over the choice of $\Omega$,*

$$\|P_\Omega(W) - pW\| \leqslant \varepsilon p\|W\|_F \,.$$

*where $\varepsilon = O(\nu\sqrt{\log d/(pd)})$.*

*Proof.* Without loss of generality we assume $\|W\|_F = 1$. The proof follows simply from application of Bernstein inequality. We view $P_\Omega(W)$ as

$$P_\Omega(W) = \sum_{i,j \in [d]^2} s_{ij} W_{ij} \delta_{ij}$$

where $\delta_{ij} \in \mathbb{R}^{d \times d}$ is the indicator matrix for entry $(i,j)$, and $s_{ij} \in \{0, 1\}$ are independent Bernoulli variable with probability $p$ of being 1. Then we have that $\mathbb{E}[P_\Omega(W)] = pW$ and $\|s_{ij}W_{ij}\delta_{ij}\| \leqslant \frac{\nu}{d}\|W\|_F$. Moreover, we compute the variance by

$$\sum_{i,j \in [d]^2} \mathbb{E}[s_{ij} W_{ij}^2 \delta_{ij}^\top \delta_{ij}] = \sum_{i,j \in [d]^2} \mathbb{E}[s_{ij} W_{ij}^2 \delta_{jj}]$$

$$= \sum_{j \in [d]} p \left( \sum_{i \in d} W_{ij}^2 \right) \delta_{jj} \qquad (D.2)$$

Therefore

$$\left\| \sum_{i,j \in [d]^2} \mathbb{E}[s_{ij} W_{ij}^2 \delta_{ij}^\top \delta_{ij}] \right\| \leqslant \frac{p\nu^2}{d}$$

Similarly we can control $\left\| \sum_{i,j \in [d]^2} \mathbb{E}[s_{ij} W_{ij}^2 \delta_{ij} \delta_{ij}^\top] \right\|$ by $p\nu^2/d$ (again notice that although $\delta_{i,j} = \delta_{j,i}$ the bounds here are correct up to constant factors). Then it follows from non-commutative Bernstein inequality [Imb10] that

$$\mathbb{P}_\Omega \left[\|P_\Omega(W) - p(W)\| \geqslant \varepsilon p\right] \leqslant d \exp(-2\varepsilon^2 pd/\nu^2) \,.$$

$\square$

**Concentration Lemmas for Noise Matrix $N$.** Next we will state the concentration lemmas that are necessary when observed matrix is perturbed by Gaussian noise. The proof of these Lemmas are really exactly the same (in fact even simpler) than the corresponding Theorem that we have just proven. The first Lemma is used in the same settings as Theorem D.1.

**Lemma D.5.** *Let $N$ be a random matrix with independent Gaussian entries $N(0, \sigma^2)$. With high probability over the support $\Omega$ and the Gaussian $N$, for any low rank matrix $W$, we have*

$$|\langle P_\Omega(N), P_\Omega(W) \rangle| \leqslant O(\sigma|Z|_\infty dr \log d + \sqrt{pd^2 r \sigma^2 |W|_\infty \|W\|_F \log d}$$

*Proof.* The proof is exactly the same as Theorem D.1 as $|\langle P_\Omega(N), P_\Omega(W) \rangle|$ is a sum of independent entries that follows from the same Bernstein's inequality. $\square$

Next we show that random sampling entries of a Gaussian matrix gives a matrix with low spectral norm.

**Lemma D.6.** *Let $N$ be a random Gaussian matrix with independent Gaussian entries $N(0, \sigma^2)$, with high probability over the choice of $\Omega$ and $N$, we have*

$$\|P_\Omega(N)\| \leqslant \varepsilon p \sigma d,$$

*where $\varepsilon = O(\sqrt{\log d/pd})$.*

*Proof.* Again the proof follows from the same argument as Lemma D.4. $\square$

# E   Auxiliary Lemmas

**Proposition E.1.** *Let $a_1, \ldots, a_r \geqslant 0$, $C \geqslant 0$. Then $C^4(a_1^2 + \cdots + a_r^2) \geqslant a_1^6 + \cdots + a_r^6$ implies that $a_1^2 + \cdots + a_r^2 \leqslant C^2 r$ and that $\max a_i \leqslant Cr^{1/6}$.*

*Proof.* By Cauchy-Schwarz inequality, we have,

$$\left( \sum_{i=1}^{r} a_i^2 \right) \left( \sum_{i=1}^{r} a_i^6 \right) \geqslant \left( \sum_{i=1}^{r} a_i^4 \right)^2 \geqslant \left( \frac{1}{r} \left( \sum_{i=1}^{r} a_i^2 \right)^2 \right)^2$$

Using the assumption and equation above we have that $a_1^2 + \cdots + a_r^2 \leqslant C^2 r$. This implies with the condition that $a_1^6 + \cdots + a_r^6 \leqslant C^6 r$, which implis that $\max a_i \leqslant Cr^{1/6}$. $\qquad\square$

**Proposition E.2.** *For any $\zeta \in (0, 1)$, there is a set $\Gamma$ of rank $r$ $d \times d$ matrices, such that for any rank $r$ $d \times d$ matrix $X$ with Frobenius norm at most 1, there is a matrix $\hat{X} \in \Gamma$ with $\|X - \hat{X}\|_F \leqslant \zeta$. The size of $\Gamma$ is bounded by $(d/\zeta)^{O(dr)}$.*

*Proof.* Standard construction of $\varepsilon$-net shows that there is a set $P \subset \mathbb{R}^d$ of size $(d/\varepsilon)^{O(d)}$ such that for any $\|u\| \leqslant 1$, there is a $\hat{u} \in P$ such that $\|u - \hat{u}\| \leqslant \varepsilon$. Such construction can also be applied to matrices and Frobenius norm as that is the same as vectors and $\ell_2$ norm.

Here we let $\varepsilon = 0.1\zeta$, and construct three sets $P_1, P_2, P_3$ where $P_1$ is an $\varepsilon$-net for $d \times r$ matrices with Frobenius norm at most $\sqrt{r}$, $P_2$ is an $\varepsilon$-net for $r \times r$ diagonal matrices whose Frobenius norm is bounded by 1, and $P_3$ is an $\varepsilon$-net for $r \times d$ matrices with Frobenius norm at most $\sqrt{r}$.

Now we define $\Gamma = \{\hat{U}\hat{D}\hat{V} | \hat{U} \in P_1, \hat{D} \in P_2, \hat{V} \in P_3\}$. Clearly the size of $\Gamma$ is as promised. For any rank $r$ $d \times d$ matrix $X$, suppose its Singular Value Decomposition is $UDV$, we can find $\hat{U} \in P_1$, $\hat{D} \in P_2$ and $\hat{V} \in P_3$ that are $\varepsilon$ close to $U, D, V$ respectively. Therefore $\hat{U}\hat{D}\hat{V} \in \Gamma$ and it is easy to check

$$\|UDV - \hat{U}\hat{D}\hat{V}\|_F \leqslant 8\varepsilon \leqslant \zeta.$$

$\qquad\square$