[Reviews · NeurIPS 2016]

Reviewer 1

Summary

The paper studies the conditions, under which the critical points of a certain regularised problem (with a quaternary term in the regularisation) coincide with optima of the matrix completion problem.

Qualitative Assessment

This is the best paper I have reviewed this year. It provides a clear exposition of an important result, linking it to relevant recent work. I would be curious if the results extended to some ensemble with uniformly distributed entries (e.g., http://arxiv.org/abs/1408.2467). The paper is very closely related to a recent work: http://arxiv.org/abs/1605.08101 but considering the submission deadline coincided with it, I doubt there can be an issue. It should like to cite it, though. Other than that, I have only trivial comments: -- F in Frobenious should be capitalised -- "our attention" (l. 229) -- "nice properties that ensure" needs no "s" (l. 251)

Confidence in this Review

2-Confident (read it all; understood it all reasonably well)


Reviewer 2

Summary

The submission studies matrix completion. A number of non-convex optimization methods for this problem are popular and perform well in practice, and there has been recent progress on showing that various methods in this class converge to a global optimum if initialization is performed carefully. The goal of the submission is to show that for a regularized version of the problem, all local minima are global minima. Hence, any optimization method that converges to a local minima will converge to a global one. Note that a number of popular algorithms, such as SGD, satisfy this property when initialized with a random or arbitrary initialization point. The results in the submission are also robust to noise. Results in this vein are known for a handful of other problems, such as dictionary learning, but these problems differ qualitatively from matrix completion. A quick summary of the (mild) limitations of the result: the proof considers specifically symmetric matrices, leaving the asymmetric case as an open question. It is also specific to the case that the objective function to be minimized is Frobenius distance from the observed matrix entries. The result makes the (apparently standard) assumptions that the condition number of the matrix is bounded, and no row makes up too big a fraction of its Frobenius norm. The analysis is interesting. It uses first- and second-order optimality conditions that any local minimum x must satisfy, to conclude that x has to equal a global minimum. In more detail, the second-order condition ensures that x has large L_2 norm. The first order condition ensures that x has small L_{infty} norm. And if x has small L_{infty} norm and large L_2 norm, then the first-order condition also ensures that x is (close to) a global minimum

Qualitative Assessment

This is a nice contribution to an important and well-studied problem. The result gives some theoretical justification for the empirical fact that popular non-convex methods for matrix completion seem to do well in practice, even when not much effort is put into initialization. The analysis is interesting and addresses specific properties of matrix completion. I think the submission is clearly above the bar for NIPS. One issue that could benefit from additional discussion: exactly what properties of the regularizer does the analysis require/exploit? And how does the proposed regularizer compare to those actually used in practice?

Confidence in this Review

2-Confident (read it all; understood it all reasonably well)


Reviewer 3

Summary

This paper considers the Burer-Monteiro factorization approach to the matrix completion problem. This paper shows that this formulation, albeit non-convex, has no bad local minimum. In fact, all spurious stationary points are shown to be strict saddle, so many optimization algorithms can escape from them and converge to the desired solution. These results are shown by analyzing the geometry of the objective function outside the strongly convex region using concentration inequalities.

Qualitative Assessment

This is solid paper on a timely topic. Several recent works show local convergence of the non-convex factorization approach to matrix completion, and this paper strengthens these results to prove global convergence. Different from several concurrent results along this line, the proof in this paper requires several new ingredients to handle the incoherence issues and the effect of the regularizer. The proof looks solid and the presentation is clear. A minor comment: The first inequality in eq. (3.9) does not seem to hold uniformly over B. For example, if x = e_1 / sqrt(d), then x is in B, the the LHS of (3.9) is p||x||^3 as Omega will miss the only non-zero entry of xx^T with high probability, but the RHS of the first inequality of (3.9) is p epsilon ||x||^3. The issue seems to be that Theorem D.2 doesn't apply to all x in B, but only to those whose infinity norm is 1/sqrt(d) smaller than its Frobenius norm. I believe this bug is easily fixable though, as Lemma 3.4 guarantees that all x of interest has Frobenius norm larger than 1/4 and hence much larger than its infinity norm, and Theorem D.2 can indeed be applied.

Confidence in this Review

3-Expert (read the paper in detail, know the area, quite certain of my opinion)


Reviewer 4

Summary

This paper proves that the commonly used non-convex objective function for matrix completion has no spurious local minima. Hence, gradient descent approaches can achieve global optimal. This result seems to be quite strong.

Qualitative Assessment

The theoretical analysis is rigorous. I believe it may lead to an important step in matrix completion field.

Confidence in this Review

2-Confident (read it all; understood it all reasonably well)


Reviewer 5

Summary

This paper discusses the geometrical properties of a nonconvex objective for completing a PSD matrix. The main theorem (Thm 2.3) says all data points x s.t grad(x) = 0 && hessian(x) >= -tau I is global minimum, this means i) All such x is a local minimum and globally optimal ii) All saddle points will have hessian(x) < -tau I (Later analysis shows tau <= 0.1p -- this implies the larger sampling probability, the large negative curvature the saddles have) This coincides with the characterization of functions with only strictly ridable saddle points in previous work, hence we know some optimization algorithms e.g. cubic regularization, trust region algorithms and SGD can converge to global optima.

Qualitative Assessment

To prove the main theorem, the authors focus on excluding critical points with (1) large coherence with standard base (e.g Lemma 3.7) and (2) large negative curvature (e.g. Lemma 3.8) from local minima. Then they show (3) the other critical points can just be the global optimums (e.g. Lemma 3.10), this boils down to showing such points satisfies Lemma 3.3 and finally C.1, where Lemma 3.3 is reached by Lemma 3.6. * part 2) I think what the authors mean in Lemma 3.8 is data points satisfying 1st and 2nd order optimality conditions will have l2 norm at least ⅛, which cannot be achieved for data points whose hessian < -0.1p I. If this is true, I would suggest that the authors rephrase Lemma 3.8 for the sake of easier reading. If not, please clarify how you exclude those points. * part 3) I think the chain in (3) is a bit long. For better presentation, I would suggest the authors use compact this part a bit. Besides, one needs to use Lemma C.1 to conclude that x satisfying Lemma 3.3 can only be equivalent solutions (e.g. z and -z for the rank 1 case). I think it’s better to include Lemma C.1 into main text as (a) this is an important step (b) Lemma C.1 requires an upper bound of epsilon -- a function of p in Lemma 3.3, the reader can see how p scales more clearly. Other small details: 1) In Thm 3.2, the authors write "tau < - 0.1p" but tau is claimed to be positive in Def 2.2. From later context (Lemma 3.4), I think the authors mean tau <= 0.1p. 2) Figure 1, the bound for l_infty norm should be 40 u / sqrt(d). This figure is also a bit confusing. The l2 ball is actually the whole ellipsoid, not a single circle. To summarize, I think this is a good paper, but the presentation could be improved. The proof idea could be outlined more clearly; how the technical definitions and intuitions are related to previous work could be included for readers not familiar with this area.

Confidence in this Review

2-Confident (read it all; understood it all reasonably well)


Reviewer 6

Summary

The authors prove that all local minimum must be global minimum for a noiseless symmetric matrix completion problem. Unlike previous works that requires good initializations, this paper shows that starting from random initialization gradient descent algorithm converges to the global minimum. Extension to noisy setting is also considered in the appendix.

Qualitative Assessment

The primary theoretical results are strong contributions and technically sound. The technical part is well-written, and all the proof strategies in the main body is easy to read. The result seems to be the first that shows all local minimums are global minimums for matrix completion problems. This opens the direction for other researchers to think about similar geometric properties in other nonconvex machine learning problems. One potential limitation is that the results only applies to the case where the completion matrix is symmetric. This restriction makes the theory less practical. That being said, I still think the results in the paper is a major step toward understanding the geometry of matrix completion problem. The paper is well-written. The organization structure is exceptional, making the paper easy to read. Although not understanding all the details in the proof, I can easily follow the proof strategies for the rank-1 case.

Confidence in this Review

1-Less confident (might not have understood significant parts)